# Insights from the Biorepository and Integrative Genomics pediatric resource

Silvia Buonaiuto [1,10], Franco Marsico[1,2,10], Akram Mohammed [3], Lokesh K. Chinthala [3], Ernestine K. Amos-Abanyie [1], Regeneron Genetics Center*, Pjotr Prins[1], Khyobeni Mozhui [1,4], Robert J. Rooney[1,5], Robert W. Williams[1,6], Robert L. Davis [3], Terri H. Finkel[7], Chester W. Brown[1,8] & Vincenza Colonna [1,2,5] ✉

The Biorepository and Integrative Genomics (BIG) Initiative in Tennessee has developed a pioneering resource to address gaps in genomic research by linking genomic, phenotypic, and environmental data from a diverse Mid-South population, including underrepresented groups. We analyzed 13,152 exomes from BIG and found significant genetic diversity, with 50% of participants inferred to have non-European or several types of admixed ancestry. Ancestry within the BIG cohort is stratified, with distinct geographic and demographic patterns, as African ancestry is more common in urban areas, while European ancestry is more common in suburban regions. We observe ancestry-specific rates of novel genetic variants, which are enriched for functional or clinical relevance. Disease prevalence analysis linked ancestry and environmental factors, showing higher odds ratios for asthma and obesity in minority groups, particularly in the urban area. Finally, we observe discrepancies between self-reported race and genetic ancestry, with related individuals self-identifying in differing racial categories. These findings underscore the limitations of race as a biomedical variable. BIG has proven to be an effective model for community-centered precision medicine. We integrated genomics education, and fostered great trust among the contributing communities. Future goals include cohort expansion, and enhanced genomic analysis, to ensure equitable healthcare outcomes.

To date, most genetic data available for human research has predominantly originated from European populations, introducing a bias in medical research and healthcare that fails to accurately represent the genetic diversity of the global human population[1–5]. Systemic inequity were aggravated by historical technological limitations such as early SNP arrays[6–8] were primarily designed based on data from European populations. Recent breakthroughs[9–14], culminating in the development of human pangenome assemblies[15,16], have finally begun dismantling these technological barriers that reinforced genetic research disparities across populations. Genetic risk assessments based on European ancestry cohorts yield less accurate outcomes for non-European populations, as seen with *CYP2C19* gene variants, which affect drug metabolism and increase risks of misdiagnosis or delayed treatment[17–19]. While the importance of including ethnically diverse populations in studies of quantitative trait evolution is well known[20], the underrepresentation of diverse populations in genetic research

[1]Dept of Genetics, Genomics and Informatics, UTHSC, Memphis, TN, USA. [2]Institute of Genetics and Biophysics, National Research Council, Naples 80111, Italy. [3]Center for Biomedical Informatics, UTHSC, Memphis, TN, USA. [4]Department of Preventive Medicine, Division of Preventive Medicine, UTHSC, Memphis, TN, USA. [5]Dept of Pediatrics, UTHSC, Memphis, TN, USA. [6]Center for Integrative and Translational Genomics, UTHSC, Memphis, TN, USA. [7]Dept of Pediatrics, Division of Rheumatology, UTHSC, Memphis, TN, USA. [8]Dept of Pediatrics, Division of Genetics, UTHSC, Memphis, TN, USA. [10]These authors contributed equally: Silvia Buonaiuto, Franco Marsico. *A list of authors and their affiliations appears at the end of the paper. ✉e-mail: vcolonna@uthsc.edu

exacerbates health inequities and limits understanding of disease genetics across ancestries, further deepening existing treatment disparities. This underrepresentation underscores the urgent need for more inclusive and diverse genetic studies to improve global health outcomes, leading to a surge of initiatives aimed at addressing these disparities (e.g., Refs. 14,21–23).

The Biorepository and Integrative Genomics (BIG) Initiative of Tennessee (US), is a multi-institute initiative that has developed a biorepository resource from a diverse Mid-South population in the US, including African Americans from Memphis - a population previously shown to have among the highest and diverse proportions of African ancestry in the United States, making it particularly valuable for studying African genetic diversity in admixed populations[24,25], and rural populations in Appalachia, which are disproportionately impacted by chronic diseases and the associated costs of healthcare[26,27]. The BIG biospecimens and their genomic data are linked to de-identified electronic health records, with the purpose of creating a platform for genomics-based research that includes underrepresented populations and to support future personalized healthcare delivery platforms[28]. The initial focus of BIG on building a large and diverse cohort for genetically informed treatment and prevention of pediatric conditions, has now been expanded to a statewide program that enrolls participants of any age with the goal of building genome-phenome-environment data for 100,000 Tennesseans.

Here we report on the analysis of 13,152 genomes from the BIG collection. We demonstrate that the BIG is a genetically diverse and ethnically rich study population, representing a unique and valuable resource for inclusive genomics. Our findings highlight ancestry-specific diversity and genetic burden, underscoring the critical need of inclusive sets of data. Finally, we show that self-reported race does not accurately reflect genetic ancestry and should be cautiously applied as a covariate in genetic analyses.

## Results

### A robust foundation for inclusive genomics studies

To date, the BIG initiative has consented over 42,000 participants with electronic health records and collected more than 15,000 biosamples from five collection sites (Fig. 1a). The BIG cohort is predominantly pediatric, with 87% of participants under 18 years old. At the time of sample collection, participant ages ranged from infancy to 90 years, with an average age of 8.4 years and a median age of 6.2 years (Supplementary Fig. 1). BIG stands out as one of the largest cohorts focused on diverse ancestries, providing a substantial representation of different ethnic backgrounds[29–35] compared to cohorts with predominantly one ancestry[36,37] (Supplementary Table 1). Notably, it is among the few cohorts specifically enriched for children with diseases, unlike most pediatric cohorts that typically recruit healthy mother-child pairs during pregnancy[30,31,33,35–40].

Since 2017, the BIG initiative has developed the Memphis Genomics Educational Network (*MEMGEN*) to engage the Memphis Shelby County public school district community in genomics education. *MEMGEN* has reached students in seven public high schools (with plans to expand to 25), providing hands-on genomic experiences and ethical discussions that inspire STEM careers and academic growth in underserved communities. Community engagement is strengthened through advisory boards like the Le Bonheur Family Partners Council, supporting the BIG initiative since 2015, and the UTHSC Community Advisory Board, representing seventeen grassroots organizations. These boards ensure research and educational efforts align with community needs, fostering a community-centered approach to precision medicine and addressing health disparities.

### Capturing broad diversity and several types of admixture

Within the BIG cohort, we identified and phased 6.8 million high-confidence variable sites, evenly distributed across the genome (Supplementary Fig. 2) through exome sequencing and genotype-by-

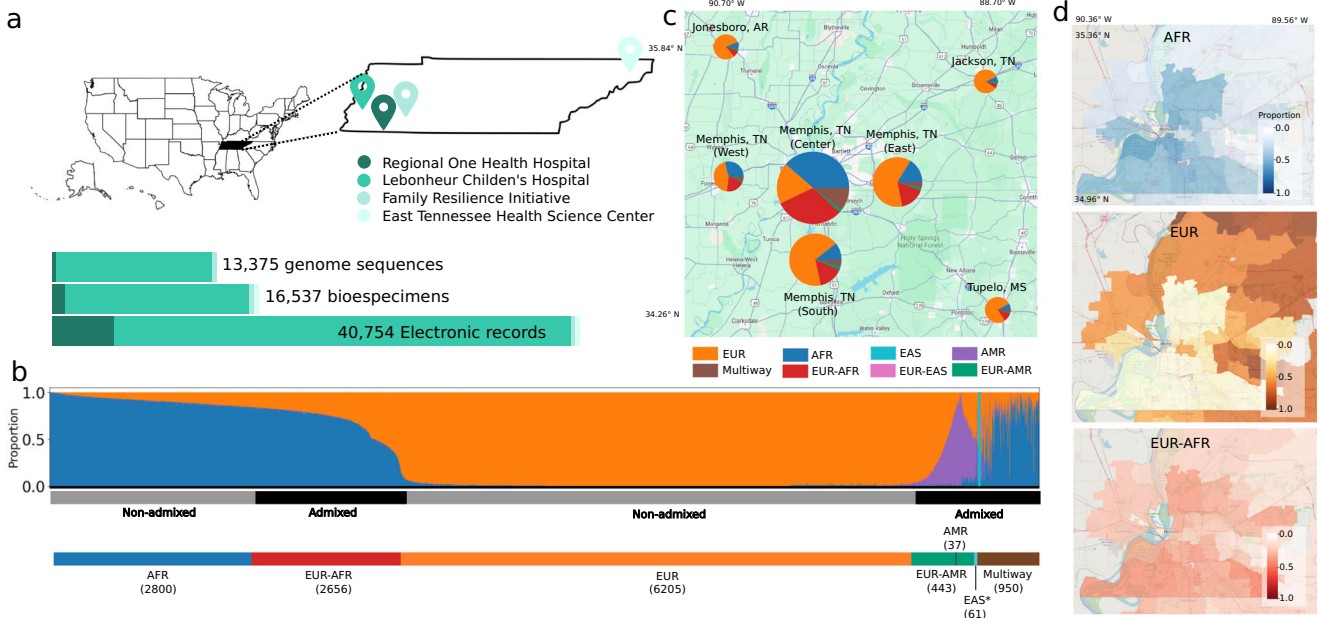

**Fig. 1 | Geographic distribution and global ancestry deconvolution of individuals from the BIG initiative. a** Overview of data collected across four sites in Tennessee, US. **b** Global ancestry deconvolution of 13,152 sequenced individuals, based on RFMix[41] and using reference populations in the 1000 Genomes and Human Genome Diversity Project (HGDP) data sets. Each vertical bar represents one individual, colors are proportional to inferred ancestry. For further analyses, individuals were grouped based on the ancestry proportions in seven categories (colored bar, number of individuals per category in parentheses), and classified as admixed or not (black and gray bar) as described in the text. **c** Proportion of individuals corresponding to each ancestry stratified by the zip code. Some colors might not be visible, see supplementary Fig. 3 or table for details. **d** Prevalence of ancestries by zip code - EUR: European; AFR: African; EAS: East-Asian; AMR: Indigenous-American. Maps were produced with the leaflet package (v. 2.2.1) using GeoJSON data for state ZIP-code boundaries publicly available.

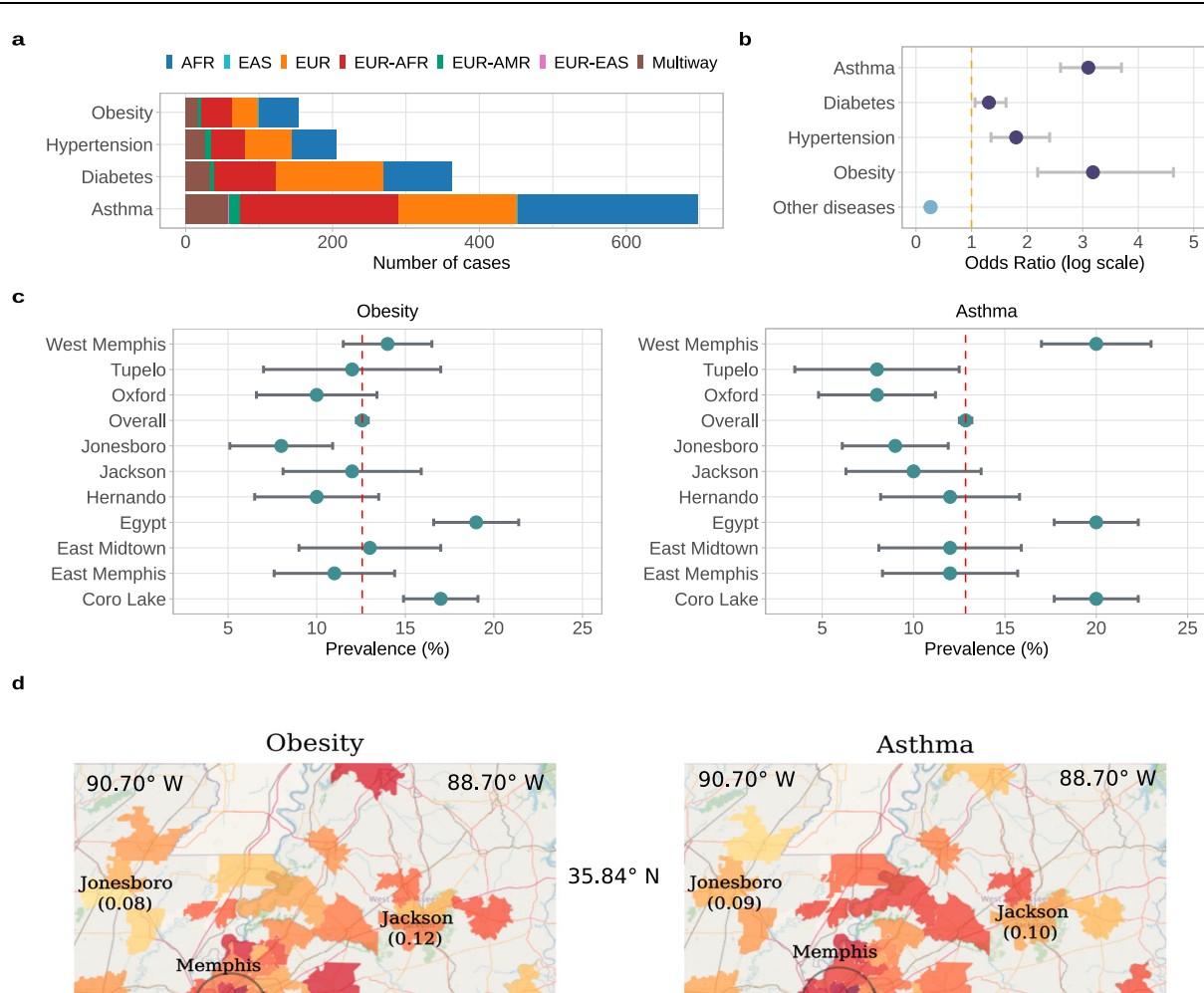

**Fig. 2 | Prevalence of diseases common in health disparities populations.**
**a** Number of cases stratified by inferred ancestry categories. **b** Odds ratios for asthma, diabetes, hypertension, and obesity compared to odds ratio of two hundred random diseases, observed among individuals self-identifying as belonging to non-White racial groups ($n = 6374$) versus White racial groups ($n = 6115$). The `Other diseases' reference represents a meta-analysis of the randomly selected diseases using the Mantel-Haenszel method. Error bars indicate 95% confidence intervals calculated using log odds ratio and its standard error. **c** Prevalence of obesity and asthma by zone. Data are presented as prevalence (proportion) with 95% confidence intervals (error bars) calculated using the Wald method. **d** The map displays zones color-coded by prevalence levels in locations with more than 100 total individuals. The Memphis Metropolitan area, characterized by high population density, is zoomed in. Maps were produced with the `leaflet` package (v. 2.2.1) using GeoJSON data for state ZIP-code boundaries publicly available.

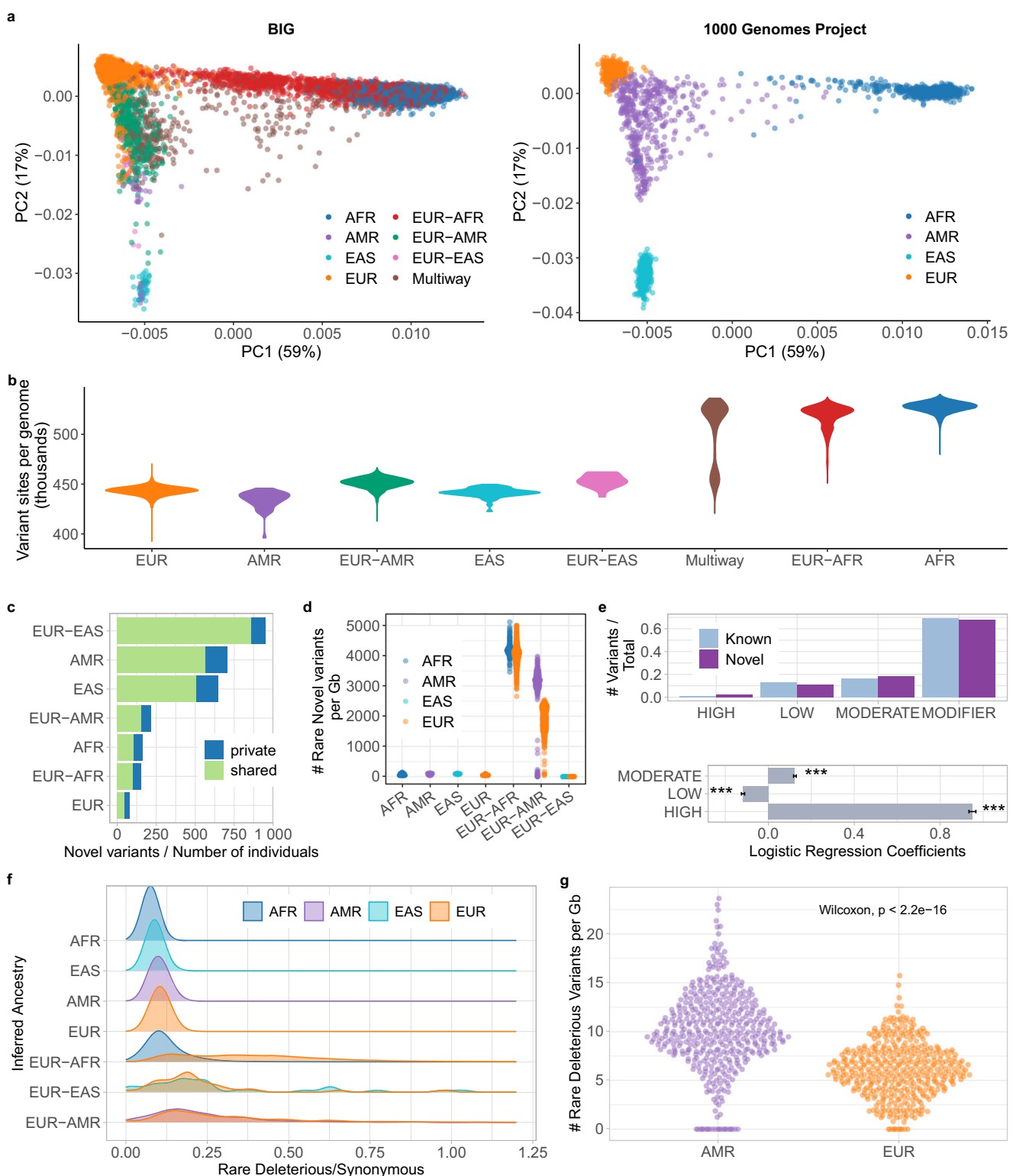

sequencing data from 13,152 individuals. We used this genetic information to understand the ancestry composition of BIG by performing supervised ancestry deconvolution[41], with 1000 Genomes and HGDP as reference populations[42,43]. While we observe a clear, uninterrupted cline of ancestry, we subdivided the data set into seven ancestry groups to account for admixture and further characterize our cohort (Fig. 1b). In practice, individuals were classified as not-admixed if more than 85% of their global ancestry corresponded to a single group. The choice of an 85% threshold reflects the understanding that genetic

ancestry exists on a continuum, therefore defining discrete categories implies setting thresholds and making arbitrary decisions (ref. 22 see Methods section). Furthermore, ancestral contributions over 10–15% are generally considered accurate and significant, while lower proportions are often linked to shorter ancestral segments and higher error rates[44].

According to this ancestry-based grouping, 50% of participants relate to individuals of non-European origin in the reference data sets. In particular, 20% of the BIG individuals are similar to Africans in the

**Fig. 3 | Genetic variability and genetic burden in the BIG cohort. a** Joint principal component analysis of genetic data from individuals in the BIG and in the 1000 Genomes populations, represented separately for clarity. Colors represent inferred genetic ancestry. The first two principal components explain 76% of the variance captured by the first 20 PCs. **b** Number of variable sites per genome compared to the reference sequence as a function of inferred ancestry. **c** Estimate of the number of novel variants by individuals per ancestry with indication of variants private to the ancestry (**d**) Count of rare novel variants by ancestry segments. Individuals in admixed groups are represented twice (**e**) Proportion of known and novel variants across different impact categories (top panel). Data are presented as ratios of variant counts to total variants, with known variants (n = 6,114,914) in light blue and novel variants (n = 771,717) in purple. The bottom panel shows logistic regression coefficients comparing the likelihood of variants being novel across impact categories, with MODIFIER serving as the reference level. Error bars represent 95%

confidence intervals. Asterisks indicate statistical significance (***p < 0.001). Detailed statistics from this logistic regression analysis are presented in Supplementary Table 3. **f** Rare deleterious-to-synonymous variant ratio across inferred ancestries. The peaks and spreads of these distributions highlight variation in the frequency of deleterious mutations across ancestries, reflecting potential differences in genetic diversity, mutation load, and evolutionary pressures. **g** Count of rare deleterious variants in EUR-AMR admixed individuals (n = 426), which have the highest deleterious-to-synonymous ratio. Variant counts are assigned based on the inferred ancestry of the genomic regions where they are found. This means individuals are counted twice: once for their AMR ancestry regions and once for their EUR ancestry regions. Statistical comparison was performed using a two-sided Wilcoxon rank-sum test with exact p-value = 2.2e-16. No adjustments were made for multiple comparisons.

reference sets, and 30% present admixed origins, with two-way and multiple-admixture patterns (Fig. 1b). The group of individual presenting more than two ancestry component is heterogeneous (Supplementary Fig. 3), consistently with previous observations[45]. These figures, projected on all consented individuals, indicate that over 20k consented samples are likely of non-European or admixed origin, placing BIG among the largest pediatric cohorts with many admixed children (Supplementary Table 1).

The distribution of inferred ancestry groups by zip code shows ancestry stratification, with prevalence of European ancestry in the suburbs and areas surrounding Memphis (Figs. 1c, 4). Stratification appears even more marked when visualized by single ancestry (Fig. 1d). A high dissimilarity index[46] between EUR and AFR (0.67) is observed, highlighting relevant geographic difference, while AFR and EUR-AFR (0.24) are the most evenly distributed pair, indicating much closer spatial overlap (Supplementary Fig. 4c). This evidence indicates that BIG individuals with similar ancestry often share a similar environment, implying that geography could act as a confounding factor if not accounted for in association analyses.

## Integrating genetic, phenotypic, and environmental information

Electronic health records are an integral part of the BIG cohort, covering a range of Phecode categories[47], with gastrointestinal and respiratory medical conditions among the most represented (Supplementary Fig. 5). We examined the prevalence of obesity, hypertension, diabetes and asthma, four health conditions commonly associated with minority groups and local environmental influences[48]. BIG children have a high incidence of diabetes and asthma (363 and 697 cases, respectively, Fig. 2a), while adults have a more balanced incidence across these same four diseases (Supplementary Fig. 6). Ancestry categories such as AFR and EUR-AFR, are major contributors across conditions, and we observed higher odds ratios for obesity and asthma in minority groups (all individuals self-identified as belonging to non-White racial groups) compared to 200 randomly selected conditions (Fig. 2b).

Analysis of disease prevalence by zip code suggests a notable environmental component for obesity and asthma. In particular, three suburban areas around Memphis exhibit above-average prevalence for both conditions, with asthma being 1.7 times more prevalent in these zones compared to the overall prevalence in BIG (≈20% versus 12.8% CI95 [12.51-13.19] Fig. 2c). While these analyses are only preliminary, the resulting observations underscore the value of the BIG dataset in linking genetic, phenotypic, and environmental information, enabling a multidimensional understanding of health disparities.

## Ancestry-specific diversity and genetic burden

Our joint principal component analysis (PCA) of the BIG and 1000 Genomes datasets (Fig. 3a, Supplementary Fig. 7) reveals significant

genetic diversity in the BIG dataset, with mixed ancestry groups contributing to the spread and overlap between clusters corresponding to African, American, East Asian, and European individuals in the 1000 Genomes. In contrast, the populations of the 1000 Genomes dataset that we used as reference for ancestry deconvolution, exhibits more distinct clustering with minimal overlap, reflecting more clearly defined ancestral groups. These results underscore the BIG dataset's value in capturing admixture and genetic diversity not represented in the 1000 Genomes, highlighting the importance of including diverse and admixed populations in genetic studies to better capture the full spectrum of human variation.

As expected, the average number of genetic differences from the reference human genome varies by ancestry[42]. Individuals with African or admixed African ancestry typically have, on average, ~85k more variable sites compared to other ancestry groups (Fig. 3b). When counting This observation underscores the risk of bias in using a single reference sequence and its associated genomic annotations. The genetic diversity represented within BIG would be more accurately modeled by a pangenomic approach[15].

Our dataset includes 771,717 novel single nucleotide variants (11.2% of the total), which are absent from major databases such as gnomAD, 1000 Genomes Project, Human Genome Diversity Project, or dbSNP[42,43,49,50]. Novel variants are mostly rare and private to ancestries, as expected (Supplementary Fig. 9). The rough number of novel variants per individual is higher within inferred admixed ancestries, Americans, and Asians (Fig. 3c). This is especially true for rare novel variants, suggesting that admixture may expose previously undetected rare variation (Fig. 3d, Supplementary Fig. 8). Some novel variants have important functional consequences on the gene product (Supplementary Fig. 9, VEP classification[51]: 2.8% high impact, including frameshift variants, stop/start gain/loss and splicing affecting variants; 19.7%: missense) and potential implications for disease association (11.0% predicted to be deleterious by SIFT[52]; 7.9% considered probably or possibly damaging by PolyPhen[53]). Notably, the rate of high impact annotation in novel variants is double compared to known variants (logistic regression coefficient $\beta = 0.95$, p-value < 0.001, Supplementary Table 3, Fig. 3e).

Genetic burden by ancestry was evaluated as the distribution of rare deleterious (alternate allele frequency <1% in the total BIG samples, predicted to have high impact or missense with SIFT<0.05 and Polyphen>0.85) versus rare synonymous genetic variants across different ancestral groups. Among non-admixed groups, African individuals display the lowest deleterious/synonymous ratio, whereas European individuals exhibit the highest (Fig. 3f). Admixed populations show broader distributions in deleterious/synonymous ratios, with the European-American group demonstrating the highest ratios. In EUR-AMR group, the average number of rare deleterious variants per Gb is significantly higher in the AMR tracts compared to EUR ones (Fig. 3g, Supplementary Fig. 10) as shown in other studies[54], likely due to demography and founder effect[55,56].

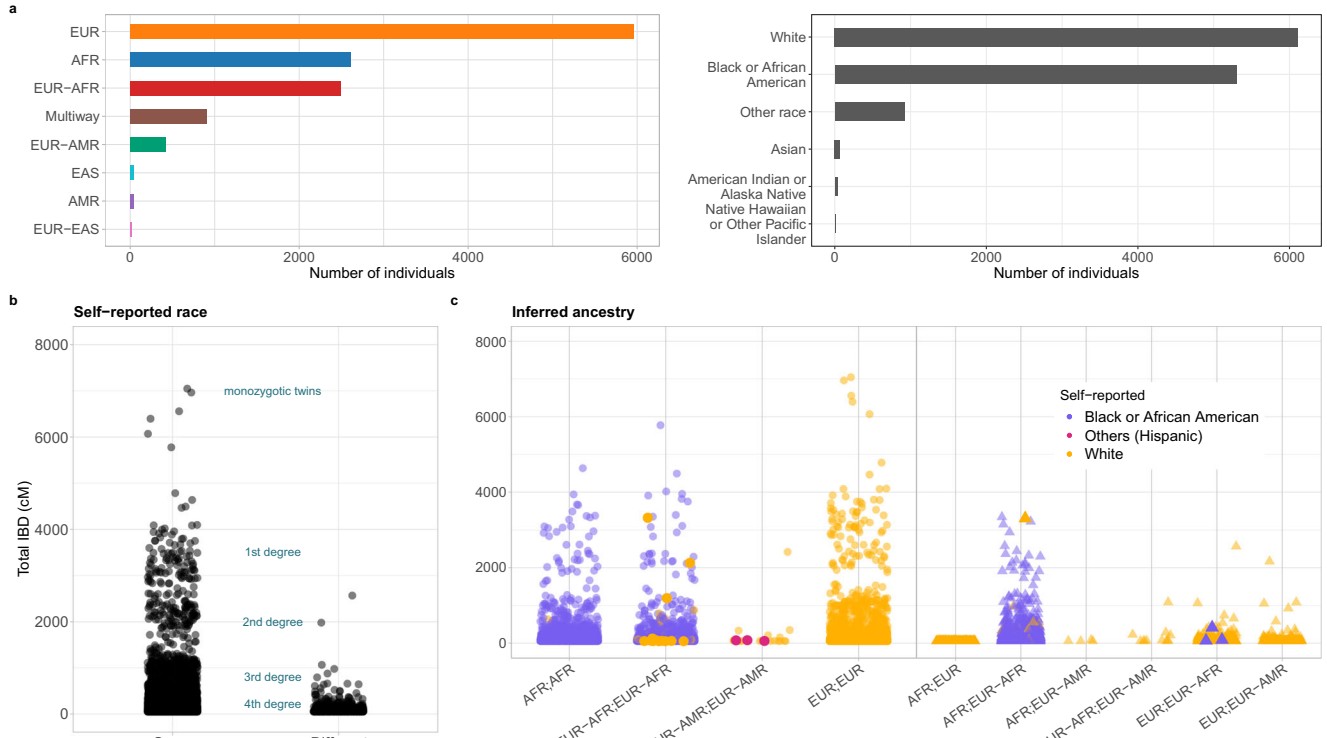

**Fig. 4 | Poor alignment between self-reported race and genetic ancestry.**
**a** Counts of individuals per inferred ancestry (left) and self-reported race (right).
**b** Genome segments shared Identical By Descent (IBD) in centimorgans (cM)
between all individual pairs in BIG, categorized by whether individuals self-reported
the same or different race. In some instances, individuals who self-report as
belonging to different races are related at the third-degree level (e.g., first cousins)
or even as close as second-degree relatives (e.g., half-siblings), as indicated by the
IBD analysis. **c** IBD genome sharing and inferred ancestry among individuals self-
reporting the same race (color-coded). In some cases, the self-reported race of a
pair deviates from the patterns observed in other pairs within the same ancestry
category.

Overall, the remarkable breadth of genetic diversity observed
underscores BIG's value as a comprehensive resource for exploring
genetic variation, enhancing disease association studies, and pro-
moting equitable genomic research in underrepresented populations.

### Discrepancies between self-reported race and inferred genetic ancestry

We compared counts of individuals in self-reported racial categories
with those in inferred genetic ancestry categories, with some racial
categories aggregated for simplicity (Supplementary Table 2). The
number of self-reported White individuals aligns closely with those
inferred as Europeans, while participants identifying as Black or Afri-
can American appear distributed between two genetic ancestry cate-
gories: Africans and admixed African-Europeans. For other racial
groups, the patterns are more diverse and complex (Fig. 4a).

We eavluated the fraction of the genome shared identical by
descent (IBD) among all possible pairs of individuals and compared
with self-reported race. Predictably, IBD genome sharing was higher
among individuals within the same self-reported race. However, we
also detected IBD sharing compatible with 2nd and 3rd degree rela-
tionships (half-siblings and 1st cousin, respectively) between indivi-
duals of different self-reported races (Fig. 4b). This observation
suggests that genetically related individuals may self-identify differ-
ently with respect to socially constructed categories like race.

The relationship between self-reported race and inferred ancestry
was further examined among pairs of individuals who identified as
belonging to the same race. In some instances, the self-reported race
of a pair differed from that of other pairs within the same ancestry
category (Fig. 4c). For example, one pair of first-degree relatives
(sharing ~50% of their genome) who both self-reported as White were
found to have differing inferred ancestries: one individual was classi-
fied as having African ancestry, while the other showed a mixture of
African and European ancestries (represented by the orange triangle in
the AFR; EUR-AFR category in Fig. 4c). Similarly, among three pairs of
individuals self-reporting as Black or African American, one member of
each pair was inferred to have European ancestry (represented by the
purple triangle in the EUR; EUR-AFR category in Fig. 4c). These findings
highlight the limitations of using self-reported race as a category for
analyzing genetic variation.

## Discussion

The BIG cohort is a genetically diverse and ethnically inclusive pedia-
tric resource, addressing the historic underrepresentation of non-
European populations in genomics research. With 87% of participants
under 18 and 50% of non-European ancestry—including 20% closely
aligning with African reference populations and 30% exhibiting com-
plex admixture patterns—it offers broad genetic variability and sig-
nificant potential to represent human genomic diversity. Previous
comparative studies have shown that admixed African populations
from Tennessee rank among those with the highest proportion of
African ancestry in the United States[25]. Notably, individuals from
Memphis exhibit the greatest genetic diversity within their African
ancestry component compared to thirteen other similar populations[24].
Although our study is not explicitly comparative, these findings posi-
tion the African and admixed African individuals in the BIG cohort as
being among the most genetically diverse populations globally simi-
larly to what observed in the highly diverse multi-ethnic biobank
BioMe[57]. The high genetic diversity observed in BIG may be associated
with the demographic and genealogical history of the African com-
ponent in Memphis, as evidenced by a recent bottleneck followed by

strong population growth[24], a line of inquiry that can be further explored in future analyses.

This diversity facilitated the discovery of new genetic variants, many of which may have clinical relevance. We have indications of ancestry-specific burden in admixed individuals. While this is an intriguing observation, it certainly deserves further investigation before any definitive conclusions can be reached. We believe that several factors, including sample size, stratification effects, and demography, must be carefully considered to achieve a more solid conclusion. This again underscores the importance of ensuring that relevant populations are well represented, as failing to do so risks leading to erroneous conclusions.

The higher number of novel variants observed in admixed individuals also deserves attention. This pattern could reflect several phenomena: First, admixture can create novel combinations of variants that were previously private to distinct ancestral populations. Second, the genetic recombination that occurs during admixture might expose previously masked deleterious variants or create new functional combinations. Third, the current reference databases may underrepresent admixed populations, making variants common in these groups appear novel in our analysis. These findings underscore both the importance of studying admixed populations and the need for more diverse reference panels in genomic research.

As a model for studying health disparities, the BIG cohort reveals higher odds ratios for obesity and asthma among minority groups, driven by genetic and environmental factors, as reflected in zip-code-specific disease patterns. We show that the BIG cohort has the potential to integrate genomic data, electronic health records, and environmental information to thoroughly analyze these and other common diseases[58]. With relevance to disease mapping, our study highlights how self-identified racial categories often fail to align with genetic ancestry, as seen in other studies[59]. The value of using race in biomedical research has been a longstanding topic of debate[60,61]. Race is predominantly a socio-cultural construct, reflecting identity and social experiences rather than genetic heritage[62]. Nevertheless, race can serve as a useful framework for describing health disparities in societies where racial categories are deeply embedded in social structures[59], and there have been increasing calls for greater inclusion of underrepresented individuals in genetic and biomedical research to help clarify the relationship between race and ancestry[63,64].

A peculiar feature of the BIG cohort is the inclusion of many admixed individuals, encompassing four distinct patterns of admixture. Admixed populations constitute a significant part of global genetic diversity and present unique statistical challenges in the analysis of genetic variation, leading to their frequent exclusion from genomics and medical research. Admixture can be used to map quantitative traits and to detect positive selection[65,66], requiring smaller sample sizes compared to other mapping techniques[67]. Admixture mapping leverages local ancestry inference to associate traits with an unusually high proportion of ancestry from one of the parental populations around the disease-causing locus[68-70] and it has been successfully used—as an example—to map Alzheimer's disease[71].

All the findings from the BIG study hold significant implications primarily for the scientific community, however, and most importantly, BIG pioneers a model for inclusive genomic studies, emphasizing community engagement to align research efforts with the needs of the contributing communities (Supplementary Fig. 11) Clinically, the insights gained from BIG can inform precision medicine initiatives for historically underserved populations, particularly in regions of Tennessee, where African Americans and others face a disproportionate burden of chronic disease. Through *MEMGEN* local students and families engage with hands-on genomics education and ethical aspects of genetic research, which demystifies the science and inspires interest in STEM fields, promoting inclusivity by respecting cultural contexts and building trust.

A future key priority for the BIG initiative is to expand its participant base to include adults, allowing for a comprehensive study across all age groups and an even broader spectrum of genetic diversity. Continued community education is also a priority to sustain engagement and participation in the BIG initiative. Another important priority is to adopt a pangenomic approach in genetic data analysis to better represent the genetic diversity within the cohort. Moving toward an inclusive genome model that integrates multiple ancestries and population-specific variants will enhance the accuracy of variant identification and genetic association studies for individuals in the BIG cohort.

By embracing this pangenomic approach, the BIG initiative can establish a benchmark for inclusive genomics, ensuring that research benefits all participants by reflecting their unique genetic backgrounds.

In conclusion, the BIG initiative can continue to lead in inclusive genomics, creating a resource that supports equitable health outcomes and advances the field toward a truly representative model of precision medicine.

## Methods

### Ethics
This study adhered to the ethical principles outlined in the Declaration of Helsinki for medical research involving human subjects. This study was conducted in accordance with ethical standards and is approved by the Institutional Review Board (IRB) of UTHSC (IRB number: 23-09204-NHSR). Written informed consent was obtained from all participants; for pediatric subjects, consent was provided by their legal guardians or next of kin. To ensure confidentiality, all data were de-identified prior to analysis.

### Sample collection sites
Le Bonheur Children's Hospital (LBCH, Memphis, TN) - LBCH is the primary pediatric care center in Memphis, and serves a predominantly African American population in an area marked by significant health disparities. Recruitment at this site was launched in October 2015 and spans inpatient rooms, ICUs, outpatient clinics, and the emergency department. The geographical provenince of enrolled individuals follow more o less a gradient that reflect distance from the hospital (Supplementary Fig. 4). Information from genomic DNA extracted from leftover blood collected during routine care is linked to de-identified electronic health record data. Leftover samples are not always available for collection, although they can be collected on a subsequent visit. This explains the discrepancy between the number of consented participants and collected biosamples.

Regional One Health (ROH, Memphis, TN)—ROH is a leading healthcare provider in Memphis, providing comprehensive care to underserved and vulnerable communities in the same geographical area of LBCH. In May 2022, the BIG Initiative extended its reach to ROH, focusing on adult genomic research. Participants are recruited across hospital settings, with DNA collected from leftover blood during standard care and linked to de-identified EHR data. This expansion complements BIG's pediatric focus at LBCH by including a diverse adult population.

East Tennessee State University (ETSU, Johnson City, TN)— The BIG Initiative expanded to ETSU in May 2023 to include the Appalachian region, emphasizing adult participant recruitment. DNA samples are collected through dedicated blood draws and linked to de-identified EHR data. ETSU's inclusion aligns with BIG's commitment to engaging rural and underserved populations, complementing efforts at LBCH and ROH to create a robust, diverse genomic database for advancing precision medicine across the Mid-South and Appalachia.

Family Resilience Initiative (FRI, Memphis, TN)—Launched in January 2019, the Family Resilience Initiative (FRI) examines the impact of

adverse childhood experiences (ACEs) and social determinants of health on long-term outcomes. The program enrolls mother-child dyads from the Memphis region, collecting sputum and/or blood samples at four visits spaced 6 months apart. Samples are processed through BIG's operational pipeline for DNA isolation, cortisol measurements, and clinical assessments. By linking biological and environmental data, FRI aims to understand ACEs' physiological and epigenetic effects, providing insights to guide tailored interventions and improve family health in vulnerable communities.

## DNA sequencing

The 13,152 samples were processed with NEB/Kapa reagents, captured with the Twist Comprehensive Exome Capture design, enhanced by Regeneron-designed spikes targeting sequencing genotyping sites. Among the sequenced samples, 95.2% achieved an average sequencing depth of at least 20X, and 99.3% of the samples had >90% of their bases covered at 20X or greater, highlighting the overall quality of the data. The genotyping spike targets an additional $\approx 1.4$ M variants in the human genome. Genotyping call rate (percentage of SNP / indels targeted genotyping at which a call can be made) is 99.0%. All samples were sequenced on an Illumina NovaSeq 6000 system on S4 flow cells sequencer using $2 \times 75$ paired-end sequencing.

## Variant identification

Sequence reads were aligned by the Burrows-Wheeler Aligner (BWA) MEM[72] to the GRCh38 assembly of the human reference genome in an alt-aware manner. Duplicates were marked using Picard, and mapped reads were sorted using sambamba[73]. DeepVariant v0.10.0 with a custom exome model was used for variant calling[74], and the GLnexus v1.2.6 tool was used for joint variant calling[75]. The variants were annotated using a Variant Effect Predictor (VEP 110)[51]. Phasing was performed using ShapeIT v5[76]. Our dataset comprised 6,886,631 variable sites after quality control, combining both exome capture and targeted sequencing data. From these sites: 135,652 variants overlapping with reference populations were used for Principal Component Analysis; 2,482,155 variants meeting RFMix filter criteria were used for Global and Local ancestry inference.

## Global and local ancestry inference

To characterize the genetic admixture within the BIG cohort, we performed a global and local ancestry inference (LAI) analysis using RFMix v.2.0; https://github.com/slowkoni/rfmix[41]. Reference samples included those of the 1000 Genomes Project and the Human Genome Diversity Project (HGDP), using the recently developed joint call[77]. The merged genotyping dataset, which combined BIG participants with reference samples, consisted of autosomal variants. To select the reference samples, we followed a quality control previously used in other studies[45]. To exclude reference samples with extensive admixture, we performed an unsupervised cluster analysis using ADMIXTURE[78]. We selected 4 groups ($k = 4$), and reference samples with a major group proportion >0.99 were considered for the analysis. Four-way LAI was performed with the number of terminal nodes for the random forest classifier set to 5 (-n 5), the average number of generations since the expected addition set to 12 (-G 12), and ten rounds of the expectation maximization algorithm (EM) (-e 10). The motivation behind the selection of $k = 4$ was our aim to characterize continental level ancestry, with four major groups: African, American, European and Asian. This aligns with the expectation for larger cities in the Americas, with the adition of the Asian group[45]. This addition was consider based on self-reported race and ethnicity categories. Reference superpopulations selected at the continental level were African (AFR), American (AMR), European (EUR), and Asian (ASN). For the ASN group, we introduced two reference populations: East Asian (EAS) and Central South Asian (CSA). CSA ancestry was negligible, with 99% of the BIG cohort showing values close to 0 and a

few cases below 0.075. As low global ancestry proportions are associated with inaccurate estimates, we excluded CSA from further analysis. Instead, we retained EAS, which showed a significant signal in a small proportion of cases consistent with the low number of individuals self-reported as Asians. Specifically, AFR is represented by YRI (101), LWK (30), MSL (16), Mbuti (10), GWD (48), ESN (64), Bantu South Africa (3), Bantu Kenya (10) and Biaka (21) groups. EUR contains Tuscan (6), Sardinian (12), Orcadian (13), IBS (117), GBR (103), French (24), Bergamo Italian (9), Basque (17) and CEU (114). AMR by Surui (6), Pima (10), PEL (10), Maya (16), Karitiana (7), and CLM (7). Finally, EAS is represented by CHS (106) and CHB (39). Local ancestry inference with RFMix2 was used to classify rare alleles (AF < 0.01), both synonymous and deleterious, by ancestry. A custom script was developed to process phased VCFs with local ancestry calls, assigning each allele to an ancestral population and generating ancestry-specific haplotype counts. This approach enables the precise tracking of allelic ancestry in samples.

Discrete ancestry categories (AMR, AFR, EUR, EAS, EUR-AMR, EUR-AFR, and Multiway) were defined based on the following criteria: (i) individuals with >85% of a single ancestry were categorized into single-ancestry groups; (ii) individuals with at least 15% contribution from two ancestries, and a combined total of over 85%, were classified as two-way admixed; (iii) individuals with significant contributions (>15%) from three or more ancestries were classified as Multiway. The 85% threshold was chosen because genetic ancestry proportion is a continuous variable, requiring arbitrary decisions when defining discrete categories (See About inferred population labels subsection), and ancestral contributions above 10–15% are generally considered accurate and significant, while lower proportions are often associated with shorter ancestral segments and higher error rates[22,44]. The number of individuals per ancestry group by ZIP code (based on ZCTA5 Code Tabulation Areas from the 2020 U.S. Census) was used to map the proportion of each ancestry within each location. The dissimilarity index[46] was calculated for ancestry categories with populations exceeding 500 individuals. To ensure reliable calculations, ZIP codes with fewer than 100 total individuals were excluded from the analysis. All geographic visualizations presented in this work were created using R. Maps were produced with the `leaflet` package (v. 2.2.1) using GeoJSON data for state ZIP-code boundaries publicly available.

## About inferred population labels

In this study, we use self-reported race and ethnicity, which are socially constructed and categorical, alongside genetic ancestry proxies derived from methods like RFMix[41]. Although race and ethnicity are discrete categories that reflect social and historical contexts, genetic ancestry arises from continuous biological processes that capture paths through the ancestral recombination graph[79]. To facilitate our analysis, we categorize genetic ancestry into regional groupings such as AMR (ancestries from the Americas) or EUR (ancestries from Europe), but it is important to clarify that these labels are not fixed or essentialized categories[80]. This grouping is useful only because it helps us explore the demographic and environmental histories that shape the variation of complex genetic traits. This discretization is merely one arbitrary scale, and in several analyses, we examine finer ancestral variation within these groupings using dimensionality reduction techniques (PCA), unsupervised clustering (ADMIXTURE) and relatedness (e.g., IBD segment analyses). We emphasize that such proxy cannot be equated with historical racial categories that have been used to justify inequality[81]. In fact, a part of the results section is focused on showing the discrepancies between both categories.

## About self-reported race

Race is self-reported by enrolled patients at the time of admission to the hospital. The admission staff select the race code from a

drop-down list of possible race categories according to HL7 standards for race and ethnicity https://hl7-definition.caristix.com/v2/HL7v2.5/Tables/0005. It is possible to select multiple race codes from the drop-down list in case people associate themselves with multiple races. Nevertheless, due to the lack of standardization in historical record collection, some of the self-reported race classifications were inaccurate or inappropriate[82]. We therefore refined the data to reflect a more reliable classification system. The criteria for refinement are detailed in Supplementary Table 2.

## Clinical data

The clinical data associated with BIG participants are extracted from the EHR (Electronic Health Records) system in flat files and shared with UTHSC through a secure file transfer protocol. These data include demographics, visits, diagnoses, procedures, prescribed and administered medications, labs, and vital signs. These data elements are converted to a limited data set (LDS) and mapped to a common data model, the OMOP (Observational Medical Outcomes Partnership) CDM. To support the analysis, the ICD9/10 diagnosis codes are assigned to PheCodes. Disease phenotypes were defined using these PheCodes: asthma was identified using Phecode RE_475; obesity using PheCodes beginning with EM_236, which includes obesity, overweight and obesity, morbid obesity, and localized adiposity; type 1 diabetes using Phecode EM_202.1; and hypertension using Phecode CV_401.

## Diversity and population structure analyses

Joint PCA, considering BIG and 1000GP cohorts, was performed in order to compare genetic diversity. We used the bigsnpr R package protocol for PCA analysis (https://privefl.github.io/bigsnpr)[83]. Briefly, this involved using King software[84] to estimate kinship coefficients and remove first and second-degree relatives (cutoff < 0.0884). LD clumping ($r < 0.2$) and exclusion of long-range LD regions were based on Mahalanobis distances. Outliers were identified with $K$-nearest-neighbor. The first 20 PCs were computed using truncated SVD. After excluding outliers, we projected related individuals in the PC space. Variants with MAF < 0.01 were excluded. For ADMIXTURE analyses, we performed unsupervised clustering with $k = 3, 4, 5$, and 6. We applied standard quality control filters, including LD pruning and removal of variants with MAF < 0.01. Logistic regression was performed in R.

## Relatedness and identical by descent analysis

We analyzed relatedness and infer family relationships using different approaches. To detect close relationships, we used firstly KING software to calculate kinship coefficients and determine the probability of sharing zero IBD (identity by descent)[84]. We also performed kinship inference using REAP, in order to account potential biases due to admixture[85]. Quality control for kinship inference included removing variants with high missingness, filtering by MAF > 0.01, and performing LD pruning.

To identify IBD segments, we used hap-ibd in the phased data set comprising 13,152 genomes, focusing on autosomal loci[86]. Hap-ibd was executed with a minimum seed parameter of 2 cM to detect IBD segments of at least this length. The inferred IBD segments were post-processed using the protocol developed by Browning et al.[87], particularly the merge-ibd-segments tool, with default parameters. Gaps with at most one discordant homozygote and <0.6 cM were removed. Total IBD between pairs of individuals was computed as the sum of the segments.

## Reporting summary

Further information on research design is available in the Nature Portfolio Reporting Summary linked to this article.

## Data availability

The data supporting this study's findings are sourced from the Biorepository and Integrative Genomics (BIG) and are not publicly available due to privacy and ethical restrictions. Access to the data is restricted to protect participant confidentiality and comply with institutional and regulatory requirements. Researchers may request access to the data after obtaining approval from the University of Tennessee Health Science Center (UTHSC) Institutional Review Board (IRB) and the BIG Research Oversight Committee. Requests should be submitted via the BIG portal at https://uthsc.edu/cbmi/big/For further assistance, please contact biglist@uthsc.edu. Data access is granted only for legitimate research purposes, and approved requestors must comply with data use agreements. Requests will typically be processed within 4 weeks of submission. Once access is granted, the data will remain available for the duration of the approved research project. We support responsible data sharing and encourage interested researchers to contact the authors or BIG for additional details.

## Code availability

The scripts used for QC, PCA, local and global ancestry deconvolution, and IBD analysis are available on https://github.com/SilviaBuonaiuto/BIG[88].

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

## Acknowledgements

We extend our gratitude to all the individuals and their families who generously contributed to the BIG initiative. We would to thank Carol Hendrix and the consent teams in Memphis and in Johnson City for oversight of recruitment and sample collection; Kito Lord, from ROH; James Adkins, and Jonathan Patrick Moorman from ETSU; Jason Yaun, Sandra Arnold from FRI; Marcella Vacca; Scott Strome; Jon McCullers; David Haines; Peter Buckley, G. Nicholas Verne, and Pamela Beckley from UTHSC; Trey Eubanks from Le Bonheur Children's Hospital; the BIG Community Advisory Board. The authors gratefully acknowledge support from the Center for Integrative and Translational Genomics at UTHSC (SB, FM, RWW, PP, VC); NIH/NIGMS (R01GM123489 to PP); NSF (PPoSS Award 2118709 to PP); the NIH/NHLBI (RO1 HL170151 to THF); The Rady Children's Institute for Genomic Medicine (THF); the Children's Foundation of Memphis (THF); the Urban Child Institute; the Children's Foundation Research Institute, Children's Foundation of Memphis; the Assisi Foundation (CWB). The Children's Research Foundation Institute, Le Bonheur Children's Hospital.

## Author contributions

SB and FM equally contributed to the study. Conceived the analyses: SB, FM, PP, AM, KM, RWW, RLD, CWB, VC; Sample sequencing: RGC; Data curation: SB, FM, AM, LKC; Formal Analysis: SB, FM, AM, EKA, VC; Funding acquisition: PP, RJR, RWW, RLD, THF, CWB, VC; Investigation: SB, FM, VC; Methodology: SB, FM, VC; Resources: PP, RWW, CWB; Software: PP; Supervision: PP, RJR, RWW, RLD, THF, CWB, VC; Visualization: SB, FM, VC; Writing - original draft: SB, FM, EKA, RWW, RLD, THF, CWB, VC; Writing - review & editing: SB, FM, AM, LKC, RGC, PP, KM, EKA, RJR, RWW, RLD, THF, CWB, VC.

## Competing interests

The Regeneron Genetic Center is a subsidiary of Regeneron Pharmaceuticals, Inc. All the other authors declare no competing interests.

## Additional information

## Regeneron Genetics Center

Aris Baras[9], Goncalo Abecasis[9], Adolfo Ferrando[9], Giovanni Coppola[9], Andrew Deubler[9], Aris Economides[9], Luca A. Lotta[9], John D. Overton[9], Jeffrey G. Reid[9], Alan Shuldiner[9], Katherine Siminovitch[9], Jason Portnoy[9], Marcus B. Jones[9], Lyndon Mitnaul[9], Alison Fenney[9], Jonathan Marchini[9], Manuel Allen Revez Ferreira[9], Maya Ghoussaini[9], Mona Nafde[9], William Salerno[9], John D. Overton[9], Christina Beechert[9], Erin Fuller[9], Laura M. Cremona[9], Eugene Kalyuskin[9], Hang Du[9], Caitlin Forsythe[9], Zhenhua Gu[9], Kristy Guevara[9], Michael Lattari[9], Alexander Lopez[9], Kia Manoochehri[9], Prathyusha Challa[9], Manasi Pradhan[9], Raymond Reynoso[9], Ricardo Schiavo[9], Maria Sotiropoulos Padilla[9], Chenggu Wang[9], Sarah E. Wolf[9], Hang Du[9], Kristy Guevara[9], Amelia Averitt[9], Nilanjana Banerjee[9], Dadong Li[9], Sameer Malhotra[9], Justin Mower[9], Mudasar Sarwar[9], Deepika Sharma[9], Sean Yu[9], Aaron Zhang[9], Muhammad Aqeel[9], Jeffrey G. Reid[9], Mona Nafde[9], Manan Goyal[9], George Mitra[9], Sanjay Sreeram[9], Rouel Lanche[9], Vrushali Mahajan[9], Sai Lakshmi Vasireddy[9], Gisu Eom[9], Krishna Pawan Punuru[9], Sujit Gokhale[9], Benjamin Sultan[9], Pooja Mule[9], Eliot Austin[9], Xiaodong Bai[9], Lance Zhang[9], Sean O'Keeffe[9], Razvan Panea[9], Evan Edelstein[9], Ayesha Rasool[9], William Salerno[9], Evan K. Maxwell[9], Boris Boutkov[9], Alexander Gorovits[9], Ju Guan[9], Lukas Habegger[9], Alicia Hawes[9], Olga Krasheninina[9], Samantha Zarate[9], Adam J. Mansfield[9], Lukas Habegger[9], Goncalo Abecasis[9], Manuel Allen Revez Ferreira[9], Joshua Backman[9], Kathy Burch[9], Adrian Campos[9], Liron Ganel[9], Sheila Gaynor[9], Benjamin Geraghty[9], Arkopravo Ghosh[9], Salvador Romero Martinez[9], Christopher Gillies[9], Lauren Gurski[9], Joseph Herman[9], Eric Jorgenson[9], Tyler Joseph[9], Michael Kessler[9], Jack Kosmicki[9], Adam Locke[9], Priyanka Nakka[9], Jonathan Marchini[9], Karl Landheer[9], Olivier Delaneau[9], Maya Ghoussaini[9], Anthony Marcketta[9], Joelle Mbatchou[9], Arden Moscati[9], Aditeya Pandey[9], Anita Pandit[9], Jonathan Ross[9], Carlo Sidore[9], Eli Stahl[9], Timothy Thornton[9], Sailaja Vedantam[9], Rujin Wang[9], Kuan-Han Wu[9], Bin Ye[9], Blair Zhang[9], Andrey Ziyatdinov[9], Yuxin Zou[9], Jingning Zhang[9], Kyoko Watanabe[9], Mira Tang[9], Frank Wendt[9], Suganthi Balasubramanian[9], Suying Bao[9], Kathie Sun[9], Chuanyi Zhang[9], Adolfo Ferrando[9], Giovanni Coppola[9], Luca A. Lotta[9], Alan Shuldiner[9], Katherine Siminovitch[9], Brian Hobbs[9], Jon Silver[9], William Palmer[9], Rita Guerreiro[9], Amit Joshi[9], Antoine Baldassari[9], Cristen Willer[9], Sarah Graham[9], Ernst Mayerhofer[9], Erola Pairo Castineira[9], Mary Haas[9], Niek Verweij[9], George Hindy[9], Jonas Bovijn[9], Tanima De[9], Parsa Akbari[9], Luanluan Sun[9], Olukayode Sosina[9], Arthur Gilly[9], Peter Dornbos[9], Juan Rodriguez-Flores[9], Moeen Riaz[9], Manav Kapoor[9], Gannie Tzoneva[9], Momodou W. Jallow[9], Anna Alkelai[9], Ariane Ayer[9], Veera Rajagopal[9], Sahar Gelfman[9], Vijay Kumar[9], Jacqueline Otto[9], Neelroop Parikshak[9], Aysegul Guvenek[9], Jose Bras[9], Silvia Alvarez[9], Jessie Brown[9], Jing He[9], Hossein Khiabanian[9], Joana Revez[9], Kimberly Skead[9], Valentina Zavala[9], Jae Soon Sul[9], Lei Chen[9], Sam Choi[9], Amy Damask[9], Nan Lin[9], Charles Paulding[9], Marcus B. Jones[9], Esteban Chen[9], Michelle G. LeBlanc[9], Jason Mighty[9], Jennifer Rico-Varela[9], Nirupama Nishtala[9], Nadia Rana[9], Jaimee Hernandez[9], Alison Fenney[9], Randi Schwartz[9], Jody Hankins[9], Anna Han[9], Samuel Hart[9], Ann Perez-Beals[9], Gina Solari[9], Johannie Rivera-Picart[9], Michelle Pagan[9] & Sunilbe Siceron[9]

[9]Regeneron Genetics Center, Tarrytown, NY, USA.

