## [Transparent Peer Review file · Nature Communications]

Insights from the Biorepository and Integrative Genomics pediatric resource

Corresponding Author: Dr Vincenza Colonna

A version of this paper was originally rejected for publication by Nature Communications, however that decision was reconsidered after appeal by the authors.

Version 0:

Reviewer comments:

Reviewer #1

(Remarks to the Author)

Buonaiuto and Marsico et al present an insightful view of the genomic, phenotypic, and environmental diversity in the BIG dataset. The authors first analyzed the genetic diversity in the dataset and reported the variation in genetic ancestry around the Memphis and surrounding geographic regions. They further studied the prevalence and distribution of multiple clinical phenotypes for different ancestry groups. The manuscript also highlights the discovery of novel variants, their consequences, and finally makes a strong case for development of similar, genetically diverse, biomedical datasets. The study is scientifically sound and would be a valuable contribution to the scientific literature. I have a few comments and questions regarding the technical analysis performed in this study:

1. Figure 1B shows an informative admixture plot but it's unclear whether the tool ADMIXTURE or RFMix was used to create the plot.
2. Is there a reason why $k=4$ (line 338-339 in the methods section) was used for defining the reference panel. Authors should justify why East Asian samples (from 1000 Genomes and HGDP) were included but South Asian samples were not in the reference panel.
3. Methods section "Clinical data" should go in finer details. For example, how were the disease (such as Asthma, Hypertension etc.) statuses defined? If Phecodes were used, then which Phecodes? Was obesity status defined using participant' height/weight data or using Phecodes?
 - a. As a small addition, it would be interesting to quantitatively compare EHR records between rural and urban zip codes and see if they vary across ancestry groups. Healthcare access and quality could vary between suburban/rural and urban geographical regions.
4. Is there a specific reason the cutoff for single-ancestry group at 85% (line 353-358 in the methods section)? It would be nice if the authors comment if they expect the rest of the 15% ancestry component as noise. If not, they should consider increasing the cutoff to a higher number.
 - a. This has implications for other results. For example, participants in AFR group resulting from a 85% cutoff would still have European haplotypes in their genome. What makes them different from EUR-AFR individuals?
5. It would be helpful for the readers to know (either through a flowchart or table) the number of markers (variants) that are used for different analyses such as QC, local ancestry inference, PCA, annotation etc.
6. For results shown in Figure 3, it might be interesting to see if the proportion of novel, known, and rare variants change when individuals from admixed genomes are considered as-is vs when European haplotypes are masked/hidden. For example, variant sites per genome (Fig. 3B) can differ for an AMR individual, depending on whether European haplotypes from that individual are considered or not. Given that the authors have already inferred local ancestry on their dataset, this could be a useful addition to the manuscript.
7. Given the availability of geographical and EHR data, it might be interesting to contrast the prevalence of Obesity and

Asthma between suburban/rural and urban populations across ancestry groups. This could be a valuable addition to Figure 2 and could provide insights into geographical & ancestral variation in health outcomes across the Memphis and surrounding areas.

Reviewer #2

(Remarks to the Author)

Buonaiuto and colleagues describe a predominantly pediatric cohort that includes African American and Appalachian participants from the Mid-South region of the US. The study includes 13,152 samples. The manuscript describes standard genetic analyses to characterize the cohort.

Lines 20-21: The Abstract states that genomes were sequenced, but the Methods indicate that an exome capture kit was used. Even with targeting of an additional 1.4M variants, I would not describe the sequences as genomes.

Lines 34-55: The tone of the first two paragraphs is angry, to the extent that if I were a reader rather than a reviewer, I would be strongly tempted to stop reading. Furthermore, the sequence data were aligned to GRCh38 following standard practices, despite the authors' criticisms. It should be noted that the human reference sequence represented by GRCh38 was constructed from multiple individuals, with ~70% of the libraries derived from an African American, and analysis of the genome-wide ancestry of the human reference sequence revealed ~50% African ancestry. Neither paragraph works in service of the manuscript or the cohort, so I recommend that the Introduction be revised to focus more on the cohort and that the first two paragraphs be eliminated.

Lines 111-117: The word continuum is misused. A continuum is a continuous sequence in which adjacent elements are imperceptibly different. In genetics, two samples are not different only if the proportion of IBD2 is 1, which occurs for duplicate samples from one individual and for monozygotic twins. In all other instances, two samples are readily distinguishable. It is accurate to describe genetic diversity in terms of gradients.

Lines 113-119: There does not appear to be a justification for forcing a binary classification of "admixed" and "non-admixed". None of the analyses, including cross-classification of ancestry with race or ethnicity, require discretizing the data in this manner. The authors admit that the threshold is arbitrary. A better approach is to estimate standard errors on the mixture proportions and perform hypothesis testing to formally assess whether a mixture proportion is significantly greater than zero. In Figure 1B, the black and gray bar can be eliminated. It is not apparent why most of the "multi-way" admixed individuals are called such, as I can only see African and European ancestry for most. In Figure 1C, I see five of the eight colors.

Lines 126-133: This is the one part of the manuscript that might move science forward. In the field of admixture mapping, researchers have recognized the possibility of confounding due to the form of stratification described by the authors. However, compelling evidence from real data has been lacking. The manuscript would be more compelling if the authors followed up on the last sentence with a concrete example.

Lines 153-160: I commend the authors for including the percent of variance explained by the PCs in Figure 3A. However, the authors need to show more than two PCs. Three PCs might be sufficient, since the authors are depicting four groups. More importantly, I do not understand how PC 1 explains 59% of the variance in the data. Based on Line 397 and a statement in the legend to Figure 3, is the denominator the proportion of variance only captured by the top 20 PCs (which is an arbitrary number), rather than the variance of the entire data set? If so, the data are so severely truncated as to render the percentages in the axis labels uninformative. I also do not endorse the protocol of Privé et al. as best practice. Decorrelating the data twice, by removing related individuals as well as by removing markers via pruning for LD, before decorrelating the data (either by PCA or SVD) is not good practice (lines 394-400). Also, excluding regions that are informative for population structure in analysis of population structure is not good practice (lines 395-396). The right figure in Panel A does not include ACB or ASW individuals (lines 342-348), who would occupy much of the space along PC 1. Inclusion of those individuals, and indeed the full set of individuals in both reference projects, would make it clear that the cohort is not capturing nearly as much novel genetic diversity as claimed (lines 156-160). To be clear, the exclusion of admixed individuals is justified for the purpose of creating reference panels for RFMix (lines 337-339).

Lines 167-169: Are the novel variants absent from the current version of dbSNP? If so, were the novel variants submitted to dbSNP?

Lines 175-177: This sentence is incomplete.

Lines 187-189: Do "EUR-AMR" individuals identify as "Hispanic"? Do they identify as "Native American" (e.g., Chickasaw)?

Lines 199-201: I am confused by the word greater. Do the authors mean 2nd degree or closer? Also, first cousins are not 2nd degree relatives (they are 3rd degree relatives), but half siblings are. Among 2nd degree relatives, half siblings belong to the same generation, whereas grandparent-grandchild relatives and aunt/uncle-niece/nephew relatives do not. Assuming that the authors have access to participant age, can the authors make a statement as to the most likely relative type? Given a predominantly pediatric cohort, 2nd degree relatives may be half siblings.

Line 227: It has not been established by this study that any variants have clinical relevance. It should not be assumed that functional consequences or "high impact" equates with clinical relevance.

Lines 246-247: It is unclear what the authors mean by “three distinct types of admixture”.

Line 317: The word coverage should be depth. Also, to what do 95.2%, 99.3%, and 90% refer?

Lines 320-321: I do not understand this sentence.

Lines 342-343: Why were South Asians excluded? Also, it is a good idea to mention that the reference Africans are all sub-Saharan and to briefly note why a Middle Eastern and North African reference is not used.

Lines 365-368: Reference 80 does not define ancestry correctly. The correct definition of ancestry is the population of origin of an allele (PMID 8981962, 9634509, and 11246470). This definition has been used without confusion by researchers investigating admixture and admixture mapping since the 1990s, and has a long history grounded in experimental crosses. Inheritance is particulate, as reported by Mendel (1866), and genetic variation is discrete. At the simplest level, the parent from which an allele is inherited is binary. The extent to which race, ethnicity, and ancestry yield concordant cross-classification does not depend on whether any of these variables is continuous.

Lines 394-395 and 403-404: The KING estimator does not account for inbreeding. I recommend that the authors investigate inbreeding using a different estimator of relatedness. Also, there might be assortative mating.

Figure 3: There is no mention of panels E and F in the Results. I am also struggling with the final two sentences in the legend to Figure 3. If variant counts are assigned based on the inferred locus-specific ancestry, then each haplotype should be counted once, and double-counting of individuals should not be an issue (lines 350-351).

Figures S4, S5, and S6: I think it would be easier to readers to discern health disparities if the figures showed prevalences rather than numbers of cases. Also, readers tend to have a difficult time with stacked bars; I recommend clustering over stacking.

Table S2: This table is highly problematic, both in terms of what “original categories” are grouped and in terms of the labels of the “grouped categories”. The stated purposes are to simplify the analyses and to eliminate inaccurate or inappropriate terminology. Given the complexity of the issues and data involved, the first stated purpose does not seem to be the way forward. The second stated purpose was not achieved.

Table S3: The title and the content do not match. There is no variable in the table that corresponds to prevalence. It would help if the model for the logistic regression was shown, so that readers could see the independent and dependent variables.

Reviewer #3

(Remarks to the Author)

The Biorepository and Integrative Genomics resource for inclusive genomics: insights from a diverse pediatric and admixed cohort by Buonaiuto et al discusses the biorepository and analysis of the 13,152 genomes within it. This is a wonderful read for the research community at all levels (basic and biomedical researchers, STEM education researchers, trainees, students, communities underrepresented in research, and lay community). Precision medicine may increase health disparities since most information is based on studies of people with European ancestry. The approach the authors took for this study to develop BIG offers the promise of precision medicine to ALL to connect ancestral foundations, improve pattern recognition to enable early interventions. The purpose of the paper is stated on line 72-75 – the author states The BIG biospecimens and their genomic data are linked to de-identified electronic health records, with the purpose of creating a platform for genomics-based research that includes underrepresented populations and to support future personalized healthcare delivery platforms.

Noteworthy results:

- Observed ancestry specific rates of novel genetic variants which are enriched for functional or clinical relevance.
- Disease prevalence analysis linked ancestry and environmental factors-showing higher odds ratios for asthma and obesity in minority groups
- Good example - limitations of race as a biomedical variable -- observed discrepancies between self-reported race and genetic ancestry, with related individuals self-identifying in differing racial categories
- An effective model for community centered precision medicine

This work is significant to the field and related fields. Using the pangenome as the reference and couple of other important tools, and the National Academies recommendations provided a clear representation of the ancestry groups studied.

I have a few minor comments listed below:

In the discussion: I'd like to see a sentence or two explaining the novel variants shown in SFig 6 in the discussion. I'm not recommending new experiments but can there be an explanation about the novel variants that appear in the admixed population. I'm curious if genetic admixture influenced the novel variants. Discuss why some novel variants appear in this study but not in other databases with African Americans. Consider if the homogeneity of the population being studied contributes to the identification of novel variants.

Did authors investigate if rare variants are consistent across different ancestries? Compare rare variants in populations with

pure African ancestry like H3 Africa versus those with African/European admixture. Determine if these variants are specific to admixed populations or if they also appear in less admixed African populations.

Line 223-226: I think this is a very important point; I think the authors should add a sentence to the Introduction explaining their choice of this population given its uniqueness as the least admixed group in the US.

Line 255 I would like to see a little more detail about the BIG community engagement. I recognize that story is probably being prepared for it's own publication but it's a good place to share for researchers that inspire to work with communities but are not sure how to get started. Basic researchers are especially challenged when trying to determine how community engagement would look like with their research questions. Maybe a flowchart in the supplement with key points and stakeholders.

Methods

Line 277: For the novice emerging genomics researcher (@ undergraduate level in particular) and STEM faculty developing CURES (Course-based Undergraduate Research Experiences), it would be nice to have more detailed explanation of the methods used, or maybe provide a Supplemental script to help others reproduce the results.

Line 289: this may not go here but I didn't understand if the patients visiting the children's hospital lived near it especially since data was collected in the emergency department. I'm guessing not that they had to travel some distance to get there. Maybe a sentence using census data (or city data) to say what the difference was in the distribution of people who come to the hospital than those that live in that area. They are coming to the hospital for some reason which is great for your study but if they didn't you wouldn't have this population in your research. It would also support coming to ER because they don't have primary care.

There were a few incomplete references:

Burchard, Esteban González ; Ziv, Elad ; Coyle, Natasha ; Gomez, Scarlett Lin ; Tang, Hua ; Karter, Andrew J ; Mountain, Joanna L ; Pérez-Stable, Eliseo J ; Sheppard, Dean ; Risch, Neil
The New England journal of medicine, 2003-03, Vol.348 (12), p.1170-1175

National Academies of Sciences, Engineering, and Medicine. 2023. Using population descriptors in genetics and genomics research: A new framework for an evolving field. Washington, DC: The National Academies Press.
<https://doi.org/10.17226/26902>.

Burchard, Esteban González ; Ziv, Elad ; Coyle, Natasha ; Gomez, Scarlett Lin ; Tang, Hua ; Karter, Andrew J ; Mountain, Joanna L ; Pérez-Stable, Eliseo J ; Sheppard, Dean ; Risch, Neil
The New England journal of medicine, 2003-03, Vol.348 (12), p.1170-1175

Reviewer #4

(Remarks to the Author)

Version 1:

Reviewer comments:

Reviewer #1

(Remarks to the Author)

I want to thank and congratulate the authors for their thorough responses to our suggestions and the revisions made to the manuscript. I recommend that the revised manuscript be accepted for publication.

Reviewer #2

(Remarks to the Author)

I appreciate the authors' good faith responses to all of my comments regarding the initial submission. The revised manuscript is substantially improved.

Reviewer #3

(Remarks to the Author)

I've reviewed the authors' responses to my comments, and I'm pleased with the thoroughness and expertise demonstrated in addressing my concerns. The authors provided sufficient data, clear explanations, and have effectively identified where the relevant information is

located.

Reviewer #4

(Remarks to the Author)

Responses to Reviewers' comments - manuscript NCOMMS-24-81002A-Z

Comments from Reviewers are in black
Our point-by-point responses are in blue

In the revised manuscript, we have indicated all modifications as follows::

- new or modified text appears in red
- deleted text is ~~struck through~~

Reviewer #1 (Remarks to the Author):

Buonaiuto and Marsico et al present an insightful view of the genomic, phenotypic, and environmental diversity in the BIG dataset. The authors first analyzed the genetic diversity in the dataset and reported the variation in genetic ancestry around the Memphis and surrounding geographic regions. They further studied the prevalence and distribution of multiple clinical phenotypes for different ancestry groups. The manuscript also highlights the discovery of novel variants, their consequences, and finally makes a strong case for development of similar, genetically diverse, biomedical datasets. The study is scientifically sound and would be a valuable contribution to the scientific literature. I have a few comments and questions regarding the technical analysis performed in this study:

1. Figure 1B shows an informative admixture plot but it's unclear whether the tool ADMIXTURE or RFMix was used to create the plot.

R: We used RFMix. This has been corrected in the caption of **Figure 1**

2. Is there a reason why $k=4$ (line 338-339 in the methods section) was used for defining the reference panel. Authors should justify why East Asian samples (from 1000 Genomes and HGDP) were included but South Asian samples were not in the reference panel.

R: We appreciate the observation from the reviewer. As another reviewer (*Reviewer #3 Lines 342-343*) pointed to the same issue, we respond together to both observations below (page 9). Shortly, the analysis with south Asians were performed, indicating negligible signals for this ancestry in our cohort. We added this observation in the methods section (**Lines 371-375**).

3. Methods section "Clinical data" should go in finer details. For example, how were the disease (such as Asthma, Hypertension etc.) statuses defined? If Phecodes were used, then which Phecodes? Was obesity status defined using participant' height/weight data or using Phecodes?

R: Thank you for noticing this missing information. In this revised version, we have provided all the previously omitted methodological details (**Lines 435-439**). We had initially overlooked including these details as they were obvious to us!

a. As a small addition, it would be interesting to quantitatively compare EHR records between rural and urban zip codes and see if they vary across ancestry groups. Healthcare access and quality could vary between suburban/rural and urban geographical regions.

R: We appreciate the suggestion from the reviewer. We had considered this analysis, but unfortunately we lack enough rural areas for a fair comparison. We used the U.S. Census Bureau's definition (Geography Program) to classify areas based on ZIP Code Tabulation Areas (ZCTAs)*. **Urban areas** are those where ZIP codes overlap mainly with Urbanized Areas (50,000+ people) or

Urban Clusters (2,500–50,000 people), while **rural areas** are those outside these boundaries. When analyzing the zip codes of the areas selected in **Figure 2D**, 90% (18) were classified as Urban and just 10% as rural (2), being all of them Metropolitan, based on the Census criteria. Nevertheless, we will reconsider the analysis as more data is incorporated in the BIG cohort.

**<https://www.census.gov/programs-surveys/geography/guidance/geo-areas/urban-rural.html>*

4. Is there a specific reason the cutoff for single-ancestry group at 85% (line 353-358 in the methods section)? It would be nice if the authors comment if they expect the rest of the 15% ancestry component as noise. If not, they should consider increasing the cutoff to a higher number.
R: We have previously detailed our rationale for selecting the 85% threshold in the Methods section (**Lines 392-397**), following the approach established in **reference [29]**, Sohail et al. (Nature, 622(7984):775-783, 2023). To provide additional clarity, we have expanded our explanation with a supplementary statement (**Lines 396-400**). In short: the 85% threshold was chosen to address the inherent complexity of creating discrete groups using genetic ancestry classification. Since genetic ancestry exists as a clinal variable rather than discrete categories, any classification system requires setting specific thresholds. The 85% cutoff is grounded in empirical evidence showing that ancestral contributions above 10-15% are generally reliable and accurately detected, while lower proportions typically correspond to shorter ancestral segments and higher error rates in ancestry inference. This threshold therefore provides a robust balance between classification accuracy and biological significance.

a. This has implications for other results. For example, participants in AFR group resulting from a 85% cutoff would still have European haplotypes in their genome. What makes them different from EUR-AFR individuals?

R: Thank you for this pertinent observation. While **Figure S10** is cited in a different context in the manuscript(*), it provides compelling evidence also to address this question. The figure demonstrates that even with our chosen ancestry classification threshold, we observe clear differences between African (AFR) and European-African admixed (EUR-AFR) individuals when analyzing variants in ancestry-specific genomic regions. This validates the robustness of our 85% threshold for ancestry classification. We are currently expanding this line of investigation in a dedicated project examining ancestry-specific incidence of functionally relevant variants.

****Figure S10** illustrates how rare deleterious variants are tracked separately by ancestral components in admixed individuals. By counting variants in both ancestral segments of admixed people (for example, counting European and American segments separately in EUR-AMR individuals), the analysis reveals how variant burden differs within the same person based on the ancestry of each genomic region.*

5. It would be helpful for the readers to know (either through a flowchart or table) the number of markers (variants) that are used for different analyses such as QC, local ancestry inference, PCA, annotation etc.

R: Agree, this information was missing. We updated the methods (**Lines 354-358**).

6. For results shown in Figure 3, it might be interesting to see if the proportion of novel, known, and rare variants change when individuals from admixed genomes are considered as-is vs when European haplotypes are masked/hidden. For example, variant sites per genome (Fig. 3B) can differ for an AMR individual, depending on whether European haplotypes from that individual are

considered on not. Given that the authors have already inferred local ancestry on their dataset, this could be a useful addition to the manuscript.

R: Thank you for this suggestion. We have proudly added a new panel D to Figure 3 in the main text! While **Figure S10** provides initial evidence for ancestry-specific patterns of variation (see also our response above to Reviewer #1, Question #4), performing this analysis at larger scale proved more computationally intensive than anticipated. The suggested masking approach has revealed intriguing patterns of variant distribution in admixed genomes. We have begun investigating these patterns and are committed to exploring them further in our ongoing research. Given the computational complexity and depth of analysis required, we plan to address this in a dedicated follow-up project (possibly a follow-up paper the same journal), as it extends beyond the current manuscript's scope but has considerable potential for understanding variant architecture in admixed populations. Nevertheless, we managed to have preliminary results, that we add to the revised manuscript as a proof of principle (**Figure 3D** and **Supplementary Figure 8**). In brief: we do have indication of ancestry specific patterns, especially for rare novel variants, as we comment in the Results (**Lines 180-181**) and Discussion (**Lines 248-255**) of the revised text.

7. Given the availability of geographical and EHR data, it might be interesting to contrast the prevalence of Obesity and Asthma between suburban/rural and urban populations across ancestry groups. This could be a valuable addition to Figure 2 and could provide insights into geographical & ancestral variation in health outcomes across the Memphis and surrounding areas.

R: As detailed in our response to *Reviewer #1, Question #3 (page 2)*, our current dataset has limited representation from rural areas. Should our data collection increase rural representation, we will incorporate rural/urban classification as a covariate in our analyses, where statistically appropriate. We appreciate this suggestion and will monitor the geographic distribution of our ongoing data collection.

Reviewer #2 (Remarks to the Author):

Buonaiuto and colleagues describe a predominantly pediatric cohort that includes African American and Appalachian participants from the Mid-South region of the US. The study includes 13,152 samples. The manuscript describes standard genetic analyses to characterize the cohort.

Lines 20-21: The Abstract states that genomes were sequenced, but the Methods indicate that an exome capture kit was used. Even with targeting of an additional 1.4M variants, I would not describe the sequences as genomes.

R: The text has been updated accordingly (**Line 20**)

Lines 34-55: The tone of the first two paragraphs is angry, to the extent that if I were a reader rather than a reviewer, I would be strongly tempted to stop reading. Furthermore, the sequence data were aligned to GRCh38 following standard practices, despite the authors' criticisms. It should be noted that the human reference sequence represented by GRCh38 was constructed from multiple individuals, with ~70% of the libraries derived from an African American, and analysis of the genome-wide ancestry of the human reference sequence revealed ~50% African ancestry. Neither paragraph works in service of the manuscript or the cohort, so I recommend that the Introduction be revised to focus more on the cohort and that the first two paragraphs be eliminated.

R: We understand the concerns about the tone of the opening paragraphs. Our intention was not to convey anger, but rather to present a novel perspective that goes beyond the common narrative of population disparities in genetic studies. We aimed to highlight “*an imbalance driven not only by systemic inequities but also by historical technological limitations*” as we explain in the original cover letter. We want to add to the debate a perspective on how technical considerations intersect with representation issues in genomics, an angle that has rarely been considered.

Regarding the point about GRCh38, we acknowledge its diverse composition and valuable contribution to the field. Our intent was not to criticize standard practices, but rather to explore additional approaches that could complement existing methods.

We appreciate that the current framing may not effectively serve our purpose. Furthermore, we agree that a more constructive approach would better introduce the cohort and its significance. To answer the reviewer’s objections, we have:

- Removed Lines 34-56 from the introduction
- Rephrased our discussion of how technological limitations can exacerbate systemic inequity in a new sentence (**Lines 60-64**). If the wording does not effectively convey our intended message, we are willing to remove the sentence entirely. We revised the language to better reflect our optimistic vision for addressing both technical and representational challenges in genomic research.

Lines 111-117: The word continuum is misused. A continuum is a continuous sequence in which adjacent elements are imperceptibly different. In genetics, two samples are not different only if the proportion of IBD2 is 1, which occurs for duplicate samples from one individual and for monozygotic twins. In all other instances, two samples are readily distinguishable. It is accurate to describe genetic diversity in terms of gradients.

R: We apologize for the terminology used. The term 'clinal' would be more accurate to describe genetic variation within human populations, as it better reflects the gradual changes in allele frequencies across geographic regions. This concept is well-documented in population genetics literature, notably since the work of Cavalli-Sforza *et al.*'s "The History and Geography of Human Genes" (1994) and related studies from decades ago (e.g., PMID:12493913, PMID: 15342553), which demonstrate how human genetic diversity typically follows continuous geographical patterns rather than discrete categories. We replaced 'continuum' with 'cline' (**Line 117**)

Lines 113-119: There does not appear to be a justification for forcing a binary classification of “admixed” and “non-admixed”. None of the analyses, including cross-classification of ancestry with race or ethnicity, require discretizing the data in this manner. The authors admit that the threshold is arbitrary. A better approach is to estimate standard errors on the mixture proportions and perform hypothesis testing to formally assess whether a mixture proportion is significantly greater than zero. In Figure 1B, the black and gray bar can be eliminated.

R: The distinction between admixed and non-admixed individuals was included to provide a general overview of admixture patterns in our cohort. We use a straightforward definition: individuals are classified as admixed when they show more than one ancestry component, and non-admixed when they show only one. While we acknowledge this is a simplified approach, it serves our purpose of providing a basic demographic description of the cohort and has proven useful in practice. For instance, the finding that '30% of BIG individuals are admixed' has already been successfully cited in

grant applications, demonstrating the value of this basic classification for communicating our cohort's diversity.

It is not apparent why most of the “multi-way” admixed individuals are called such, as I can only see African and European ancestry for most.

R: We appreciate your attention to the figure's visibility. We have addressed this by providing a supplementary plot with a detailed view of the multiway group (newly added **supplementary Figure 3**, also reported below). Our focus on scientific rigor occasionally outpaces our design skills, and we welcome such feedback to improve data presentation. As can be seen in the figure, the multiway group contains a significant proportion of three ancestries, as also detailed in Methods section (**Lines 396-397**). The composition of the group is highly heterogeneous, potentially indicating complex demographic histories and diversity within the group. We added this observation in the results section (**Lines 129-130**)

In Figure 1C, I see five of the eight colors.

R: As above, we understand the proportion disparity makes it difficult to see all groups in the plot. For this reason, we added a supplementary Table (**Table S4**) with exact values to ensure all ancestry components are clearly documented and accessible to readers.

Lines 126-133: This is the one part of the manuscript that might move science forward. In the field of admixture mapping, researchers have recognized the possibility of confounding due to the form of stratification described by the authors. However, compelling evidence from real data has been lacking. The manuscript would be more compelling if the authors followed up on the last sentence with a concrete example.

R: We appreciate this valuable comment highlighting the manuscript's potential contribution to admixture mapping methodology. While we agree that this observation has significant implications, we deliberately kept this section descriptive as the detailed analysis of these patterns warrants its own dedicated investigation. We are currently consolidating the results of a comprehensive study of these stratification effects. Preliminary results show promising insights

and we have already a quite advanced draft for a new manuscript. However, including these findings in this manuscript would extend beyond its scope, i.e. provide the first characterization of the BIG cohort. We look forward to sharing these results in a forthcoming publication focused specifically on admixture mapping methodology.

Lines 153-160: I commend the authors for including the percent of variance explained by the PCs in Figure 3A. However, the authors need to show more than two PCs. Three PCs might be sufficient, since the authors are depicting four groups.

R: Thank you for this suggestion for visualization. We have generated plots incorporating three principal components (see below, different views), but we found they do not provide substantial additional information beyond what is visible in the two-component visualization. We include these 3D plots (several views) here for your reference. However, we recommend maintaining the current 2D representation in the manuscript, as three-dimensional plots can often be challenging to interpret, especially in static format, and may obscure rather than clarify population relationships. We are happy to include these as supplementary material if you feel they would be valuable to readers.

Four different views of data in Figure 3A obtained plotting the first three principal components instead of only two. Color scheme is the same as in Figure 3A

More importantly, I do not understand how PC 1 explains 59% of the variance in the data. Based on Line 397 and a statement in the legend to Figure 3, is the denominator the proportion of variance only captured by the top 20 PCs (which is an arbitrary number), rather than the variance of the entire data set? If so, the data are so severely truncated as to render the percentages in the axis labels uninformative.

R: We appreciate this statistical observation. Indeed, we understand the implications of using truncated variance components and acknowledge that retaining only a subset of PCs may simplify complex genetic relationships. However, we followed established practice in human genetics to facilitate direct comparisons with existing literature. This approach of using the top 20 PCs has become standard in population genetics analyses (e.g., 1000 Genomes Project and UK Biobank analyses) and has proven effective for capturing major population structure patterns. While we could elaborate on this methodological choice in the Methods section, we opted for consistency with field standards to ensure our results remain comparable with other major genomic studies. Nevertheless, for clarity we add the scree plot of the first ten components as Supplementary Figure S7, and we report it here for simplicity.

I also do not endorse the protocol of Privé et al. as best practice. Decorrelating the data twice, by removing related individuals as well as by removing markers via pruning for LD, before decorrelating the data (either by PCA or SVD) is not good practice (lines 394-400). Also, excluding regions that are informative for population structure in analysis of population structure is not good practice (lines 395-396).

R: Thank you for this methodological comment regarding our analytical approach. While we recognize that common practice does not automatically equate to best practice, the protocol we followed (Privé et al.) has been effectively employed in numerous large-scale genomic studies, including the UK Biobank and recent pan-ancestry analyses. We acknowledge that different approaches to data decorrelation and population structure analysis exist, each with their own theoretical merits. Our choice was primarily driven by enabling comparability with existing datasets in the field. We appreciate this constructive critique and would be interested in exploring alternative frameworks in future analyses

The right figure in Panel A does not include ACB or ASW individuals (lines 342-348), who would occupy much of the space along PC 1. Inclusion of those individuals, and indeed the full set of individuals in both reference projects, would make it clear that the cohort is not capturing nearly as much novel genetic diversity as claimed (lines 156-160). To be clear, the exclusion of admixed individuals is justified for the purpose of creating reference panels for RFMix (lines 337-339).

R: We appreciate that the Reviewer identified the key rationale behind this methodological choice. As they correctly point out, excluding admixed individuals is justified when creating reference panels for RFMix. We applied this approach consistently throughout our analyses to maintain methodological coherence. The reviewer's comment precisely captures our reasoning. We have added a few words to clarify this in the Results (**Lines 164-165**) and cited in the Discussion another biobank that presents similar patterns of genetic diversity (**Lines 236-240**)

Lines 167-169: Are the novel variants absent from the current version of dbSNP? If so, were the novel variants submitted to dbSNP?

R: We are in the process of submitting the data about novel variants to dbSNP.

Lines 175-177: This sentence is incomplete.

R: Yes, there is a missing word, thanks for noticing we added the word 'variants' (**Line 187**)

Lines 187-189: Do "EUR-AMR" individuals identify as "Hispanic"? Do they identify as "Native American" (e.g., Chickasaw)?

R: We can not answer this question because we do not have the data required. We related ancestry to race (<https://hl7-definition.caristix.com/v2/HL7v2.5/Tables/0005>) not to ethnicity, this is the sole data we have available. When people self-report their race as 'Hispanic' we consider it as an incorrect response, as outlined in **Supplementary Table S2**, and they get classified as 'other race'.

Lines 199-201: I am confused by the word greater. Do the authors mean 2nd degree or closer? Also, first cousins are not 2nd degree relatives (they are 3rd degree relatives), but half siblings are. Among 2nd degree relatives, half siblings belong to the same generation, whereas grandparent-grandchild relatives and aunt/uncle-niece/nephew relatives do not. Assuming that the authors have access to participant age, can the authors make a statement as to the most likely relative type? Given a predominantly pediatric cohort, 2nd degree relatives may be half siblings.

R: Yes, it is a typos that we corrected in (**Line 209-210**).

Line 227: It has not been established by this study that any variants have clinical relevance. It should not be assumed that functional consequences or "high impact" equates with clinical relevance.

R: Yes, we agree, we soften our claims (**Line 241**)

Lines 246-247: It is unclear what the authors mean by "three distinct types of admixture".

R: Yes, it is a typo and also not clear. We replaced 'types' with 'patterns' (**Line 269**). We actually see four patterns: 1. EUR-AFR, 2. EUR -ASN, 3. EUR-AMR, and 4. Multiway and this has also been corrected (**Line 269**)

Line 317: The word coverage should be depth. Also, to what do 95.2%, 99.3%, and 90%

R: Yes, sorry for confusion, the whole sentence was a typo, we have rephrased (**Lines 341-342**)

Lines 320-321: I do not understand this sentence.

R: Yes, sorry this sentence was a typo and has been removed (**Lines 345-346**)

Lines 342-343: Why were South Asians excluded? Also, it is a good idea to mention that the reference Africans are all sub-Saharan and to briefly note why a Middle Eastern and North African reference is not used.

R: We appreciate this observation from the reviewer. Indeed, we performed an initial analysis including South Asian references, and the following results were obtained:

Proportion of South Asian ancestry in BIG cohort (HGDP-1kGP joint call reference panel, superpopulation CSA)	
South Asia Global ancestry proportion	Proportion of individuals
0-0.025	0.9872392451
0.025-0.05	0.00901658134
0.05-0.075	0.001528234125
> 0.075	0

As it can be seen, and consistent with the low proportion of individuals self-reported as Asians in BIG, we have a very small proportion of South Asian ancestry. As previously mentioned, low global ancestry is associated with lower accuracy, and adding non-significant reference groups can introduce noise and mislead the results. For this reason, we decided to exclude South Asian analysis, and also consider strong correlation between individuals self-reported as Asians and East Asian Ancestry. To further clarify, we added this explanation in the methods section (**Lines 376-381**).

Lines 365-368: Reference 80 does not define ancestry correctly. The correct definition of ancestry is the population of origin of an allele (PMID 8981962, 9634509, and 11246470). This definition has been used without confusion by researchers investigating admixture and admixture mapping since the 1990s, and has a long history grounded in experimental crosses. Inheritance is particulate, as reported by Mendel (1866), and genetic variation is discrete. At the simplest level, the parent from which an allele is inherited is binary. The extent to which race, ethnicity, and ancestry yield concordant cross-classification does not depend on whether any of these variables is continuous.

R: Our understanding is that the reviewer disagree with this sentence in the Methods (section "About inferred population labels") :

"Although race and ethnicity are discrete categories that reflect social and historical contexts, genetic ancestry arises from continuous biological processes that capture paths through the ancestral recombination graph"

That we use as an explanation of the criteria used to label individuals using ancestry following the guidelines outlined in the reference:

- Lewis, Anna CF, et al. *Science Getting genetic ancestry right for science and society*. **Science** **2022**, 376(6590):250–252

The reviewer disagree with the guidelines in this reference, and suggests instead a definition based on these three papers:

1. McKeigue PM. *Mapping genes underlying ethnic differences in disease risk by linkage disequilibrium in recently admixed populations*. **Am J Hum Genet.** **1997** Jan;60(1):188-96. PMID: 8981962; PMCID: PMC1712571.
2. McKeigue PM. *Mapping genes that underlie ethnic differences in disease risk: methods for detecting linkage in admixed populations, by conditioning on parental admixture*. **Am J Hum Genet.** **1998** Jul;63(1):241-51. doi: 10.1086/301908. PMID: 9634509; PMCID: PMC1377232.
3. McKeigue PM, Carpenter JR, Parra EJ, Shriver MD. *Estimation of admixture and detection of linkage in admixed populations by a Bayesian approach: application to African-American*

populations. **Ann Hum Genet.** 2000 Mar;64(Pt 2):171-86.
doi:10.1017/S0003480000008022. PMID: 11246470.

We have carefully read two of the three papers, we could not obtain full text of the paper #3, great if the Reviewer could help on that. To facilitate discussion, we include here the relevant sections:

1. **Paper #1.** This 1997 paper from McKeigue emphasizes the critical importance of accurate ancestry assignment at marker loci in admixture mapping, and provide several practical approaches for defining ancestry using restriction fragment length polymorphisms (RFLPs) markers. Indications are: (a) Look for marker alleles that occur in only one of the two populations; (b) Group alleles at several closely spaced marker loci into haplotypes; (c) practical considerations on marker spacing. The paper does not directly explain the step-by-step process of how to assign ancestry to individuals. While it discusses methods for identifying ancestry-informative markers (like using RFLPs, haplotypes, or representational difference analysis), it does not provide a specific methodology for using these markers to assign ancestry to individuals
2. **Paper #2.** This 1998 paper by McKeigue defines ancestry in admixed populations by introducing two key concepts. First, he develops the "marker information content for ancestry" (f), a quantitative measure that represents how much ancestry uncertainty is reduced when an allele is typed at a marker locus. For biallelic markers, f is calculated from allele frequencies in the founding populations, with a suggested threshold of 30% for marker selection. Second, he presents a quantitative framework using hidden Markov models to estimate ancestry at each locus, demonstrating how information from multiple nearby markers can be combined to more accurately assign ancestry and calculate probability distributions of ancestry states.

We appreciate the reviewer's suggestions regarding alternative methods for inferring genetic ancestry. However, our study employs a different approach that is widely accepted in the scientific community. The method we chose has been extensively validated and is particularly well-suited for analyzing current types of genetic data. We are confident in its validity for our research objectives.

We agree with the reviewer's statement that "At the simplest level, the parent from which an allele is inherited is binary." However, while this binary inheritance pattern is true for individual alleles, it does not address the more complex question of how to define overall ancestry when considering all genetic markers simultaneously. This complexity becomes particularly relevant when we need to assign ancestry-based labels to individuals.

Regarding the statement "The extent to which race, ethnicity, and ancestry yield concordant cross-classification does not depend on whether any of these variables is continuous" - while this theoretical discussion is intellectually compelling, it falls outside the scope of our current research focus.

Lines 394-395 and 403-404: The KING estimator does not account for inbreeding. I recommend that the authors investigate inbreeding using a different estimator of relatedness. Also, there might be assortative mating.

R: Thank you for this suggestion. We have also used REAP to account for admixture. It is added in the Methods section (**Line 454-455**). The relatedness inference was performed as a secondary study for PCA. Main relatedness results are associated with IBD estimation.

Figure 3: There is no mention of panels E and F in the Results. I am also struggling with the final two sentences in the legend to Figure 3. If variant counts are assigned based on the inferred locus-specific ancestry, then each haplotype should be counted once, and double-counting of individuals should not be an issue (lines 350-351).

R: Thanks for spotting this typo that we now corrected (C corrected as 'E' in **Line 192**; 'D' corrected as 'F' (**Line 195**))

Figures S4, S5, and S6: I think it would be easier to readers to discern health disparities if the figures showed prevalences rather than numbers of cases. Also, readers tend to have a difficult time with stacked bars; I recommend clustering over stacking.

R: For all the three figures, we provide below two versions (stacks and clusters). For all of them we feel that given the high number of categories and in some cases hierarchies, we found the stacked version easier to understand. However, we would accept what the editor/reviewer think would work best.

Revised version of **Figure S4** showing both stacked (original) and clustered (revised) layouts.

Revised version of **Figure S5** showing both stacked (original) and clustered (revised) layouts.

Figure S6. Revised version showing both stacked (original) and clustered (revised) layouts.

Table S2: This table is highly problematic, both in terms of what “original categories” are grouped and in terms of the labels of the “grouped categories”. The stated purposes are to simplify the analyses and to eliminate inaccurate or inappropriate terminology. Given the complexity of the issues and data involved, the first stated purpose does not seem to be the way forward. The second stated purpose was not achieved.

R: Thank you for the opportunity to improve the table and better explain our work. We think this is a great addition to the paper. We would like to clarify the motivations behind this effort. Data collection regarding race has been heterogeneous. It's important to note that this project has been collecting data for eight years through various health records information systems. While recent records were collected as described in the Methods section under "About self-reported race" (Lines 421-425) using a dropdown menu with Standardized HL7 v2.5 Race Categories, earlier record collection was more varied. This resulted in the inclusion of different terms, some of which are inappropriate (e.g., "Caucasian") or describe ethnicity rather than race (e.g., "Hispanic").

Following standard data analysis practices, we standardized all self-reported race data according to the HL7 standard. In the revised version, we made these changes:

- **Methods:** We added text explaining the heterogeneity and potential inaccuracies of self-reported race (**Lines 425-428**)
- **Table S2 Labels:** We replaced "Original Category" with "Self-Reported Race" and "Grouped Category" with "Standardized HL7 v2.5 Race Category"

Regarding the reviewer comments:

- *simplifying the analyses:* While we are fully aware of the complexity of using race as a descriptor (see **Figure 4** of our paper), we observed 21 self-reported races in our data. Due to the low frequency of some categories, we decided to aggregate them. This table describes our aggregation decisions, as analysis with seven categories is more robust than with 21 categories, some of which had very few cases (e.g. Amharic (East African)=1, Asian or Pacific Islander=1).
- *elimination of inaccurate terminology:* We maintain our position on this matter. Terms like "Caucasian" are inappropriate and scientifically inaccurate (see: Alice B Popejoy. *Too many scientists still say caucasian*. **Nature**, 596(7873):463–463, 2021), and we chose not to retain them in the text.

Table S3: The title and the content do not match. There is no variable in the table that corresponds to prevalence. It would help if the model for the logistic regression was shown, so that readers could see the independent and dependent variables.

R: Thanks for spotting this typo, we changed the text in the revised **Table S3**.

Reviewer #3 (Remarks to the Author):

The Biorepository and Integrative Genomics resource for inclusive genomics: insights from a diverse pediatric and admixed cohort by Buonaiuto et al discusses the biorepository and analysis of the 13,152 genomes within it. This is a wonderful read for the research community at all levels (basic and biomedical researchers, STEM education researchers, trainees, students, communities underrepresented in research, and lay community). Precision medicine may increase health disparities since most information is based on studies of people with European ancestry. The approach the authors took for this study to develop BIG offers the promise of precision medicine to ALL to connect ancestral foundations, improve pattern recognition to enable early interventions. The purpose of the paper is stated on line 72-75 – the author states The BIG biospecimens and their genomic data are linked to de-identified electronic health records, with the purpose of creating a platform for genomics-based research that includes underrepresented populations and to support future personalized healthcare delivery platforms.

Noteworthy results:

- Observed ancestry specific rates of novel genetic variants which are enriched for functional or clinical relevance.
- Disease prevalence analysis linked ancestry and environmental factors-showing higher odds ratios for asthma and obesity in minority groups

- Good example - limitations of race as a biomedical variable -- observed discrepancies between self-reported race and genetic ancestry, with related individuals self-identifying in differing racial categories
- An effective model for community centered precision medicine

This work is significant to the field and related fields. Using the pangenome as the reference and couple of other important tools, and the National Academies recommendations provided a clear representation of the ancestry groups studied.

I have a few minor comments listed below:

In the discussion: I'd like to see a sentence or two explaining the novel variants shown in SFig 6 in the discussion. I'm not recommending new experiments but can there be an explanation about the novel variants that appear in the admixed population. I'm curious if genetic admixture influenced the novel variants.

R: We agree. To address this question about novel variants in admixed populations, we added a paragraph in the discussion (**Lines 248-255**) that explores how genetic admixture might influence these variants through multiple mechanisms, including the creation of novel variant combinations from distinct ancestral populations, the potential unmasking of previously hidden deleterious variants through recombination, and the possibility that current reference databases underrepresent admixed populations.

Discuss why some novel variants appear in this study but not in other databases with African Americans. Consider if the homogeneity of the population being studied contributes to the identification of novel variants.

R: Great suggestion. The answer is related to the demography of these populations. As we mentioned in the discussion section, in Browning et al., *PLoS Genetics*, 2018, the authors showed that the African component of individuals from Memphis, Tennessee, exhibits the greatest genetic diversity. In that study, the effective population size analysis indicated a recent bottleneck followed by an expansion, which could explain this phenomenon.

To further investigate this, we are conducting studies on the demography and genealogy of the Memphis population. We have preliminary results that confirm the observations of Browning et al. in our cohort, and we would be happy to share the figure in this response. However, we are currently in the process of expanding the analysis with new data. As an anticipation that we prefer to keep for a separate publication, in the plot below, we show effective population size estimated based on IBD segments (using IBDNe, Browning et al., *PLoS Genetics*, 2018).

Figure: Effective Population Size (Ne) of the African component in BIG. The Y-axis shows Ne in Log10 Scale. X-axis indicates the number of generations ago from the present (generation zero). The blue line indicates the estimated Ne values, while the shaded area reflects the 95% confidence intervals, highlighting the uncertainty around the estimates.

The plot shows a recent increase in effective population size (N_e), indicating a strong demographic expansion in the last generations. This rapid growth reduces the effect of genetic drift, allowing rare variants to accumulate and persist in the population. In contrast, older generations (further back in time) show a reduced N_e , consistent with a historical bottleneck around 12 generations 300 years ago (considering 25 years per generation).

We appreciate the comment of the reviewer, and we decided to expand the discussion (**Lines 236-240**) related to the diversity of the African component in Memphis. In future studies, we will go further in this direction of characterizing the demography and genealogy of the population.

Did authors investigate if rare variants are consistent across different ancestries? Compare rare variants in populations with pure African ancestry like H3 Africa versus those with African/European admixture. Determine if these variants are specific to admixed populations or if they also appear in less admixed African populations.

R: See the response to *Reviewer 1 question #6*. We have conducted this very interesting analysis to the point of having preliminary results that we share in the revised manuscript (**Figure S8, Lines 180-181**). As we point out in the above response, these results merit more detailed exploration and, together with the demographic analyses mentioned in our previous response, would be better suited for a separate research paper. In the spirit of current paper providing a general characterization of the BIG cohort and a showcase of ongoing/possible research, we decided to include a brief overview of the preliminary results.

Line 223-226: I think this is a very important point; I think the authors should add a sentence to the Introduction explaining their choice of this population given its uniqueness as the least admixed group in the US.

R: Agree, we added a sentence and references to it in **Lines 75-77**

Line 255 I would like to see a little more detail about the BIG community engagement. I recognize that story is probably being prepared for its own publication but it's a good place to share for researchers that inspire to work with communities but are not sure how to get started. Basic researchers are especially challenged when trying to determine how community engagement

would look like with their research questions. Maybe a flowchart in the supplement with key points and stakeholders.

R: Agree, we had the material ready for this, and now we have a newly added **Supplementary Fig 10** referenced in **Line 280**.

Methods

Line 277: For the novice emerging genomics researcher (@ undergraduate level in particular) and STEM faculty developing CURES (Course-based Undergraduate Research Experiences), it would be nice to have more detailed explanation of the methods used, or maybe provide a Supplemental script to help others reproduce the results.

R: All the code is publicly available in the git repository <https://github.com/SilviaBuonaiuto/BIG> and this web link is mentioned in Methods (**Line 466**)

Line 289: this may not go here but I didn't understand if the patients visiting the children's hospital lived near it especially since data was collected in the emergency department. I'm guessing not that they had to travel some distance to get there. Maybe a sentence using census data (or city data) to say what the difference was in the distribution of people who come to the hospital than those that live in that area. They are coming to the hospital for some reason which is great for your study but if they didn't you wouldn't have this population in your research. It would also support coming to ER because they don't have primary care.

R: If we understood correctly, here the question is: *Do the patients visiting the children's hospital emergency department live nearby, or do they travel from further away?*

If this is correct, then **Figure S4** should answer, as it show where the patient lives on the geographical map. The answer to the question is that we do observe a gradient that reflect distance from the hospital, as expected. We added a sentence in the text (**Lines 311-313**) that highlights this distribution and its key implications.

There were a few incomplete references:

Burchard, Esteban González ; Ziv, Elad ; Coyle, Natasha ; Gomez, Scarlett Lin ; Tang, Hua ; Karter, Andrew J ; Mountain, Joanna L ; Pérez-Stable, Eliseo J ; Sheppard, Dean ; Risch, Neil
The New England journal of medicine, 2003-03, Vol.348 (12), p.1170-1175

National Academies of Sciences, Engineering, and Medicine. 2023. Using population descriptors in genetics and genomics research: A new framework for an evolving field. Washington, DC: The National Academies Press. <https://doi.org/10.17226/26902>.

Burchard, Esteban González ; Ziv, Elad ; Coyle, Natasha ; Gomez, Scarlett Lin ; Tang, Hua ; Karter, Andrew J ; Mountain, Joanna L ; Pérez-Stable, Eliseo J ; Sheppard, Dean ; Risch, Neil
The New England journal of medicine, 2003-03, Vol.348 (12), p.1170-1175

R: Thanks for spotting it, we have fixed the missing parts.

Reviewer #4 (Remarks to the Author):
